# Vulnerability of invasive glioblastoma cells to lysosomal membrane destabilization

Vadim Le Joncour[1],[†] (iD), Pauliina Filppu[1],[†], Maija Hyvönen[1], Minna Holopainen[2], S Pauliina Turunen[3],[4], Harri Sihto[1], Isabel Burghardt[5], Heikki Joensuu[1],[6], Olli Tynninen[7], Juha Jääskeläinen[8], Michael Weller[5], Kaisa Lehti[3],[4], Reijo Käkelä[2] & Pirjo Laakkonen[1],[9],* (iD)

## Abstract

The current clinical care of glioblastomas leaves behind invasive, radio- and chemo-resistant cells. We recently identified mammary-derived growth inhibitor (MDGI/*FABP3*) as a biomarker for invasive gliomas. Here, we demonstrate a novel function for MDGI in the maintenance of lysosomal membrane integrity, thus rendering invasive glioma cells unexpectedly vulnerable to lysosomal membrane destabilization. MDGI silencing impaired trafficking of polyunsaturated fatty acids into cells resulting in significant alterations in the lipid composition of lysosomal membranes, and subsequent death of the patient-derived glioma cells via lysosomal membrane permeabilization (LMP). In a preclinical model, treatment of glioma-bearing mice with an antihistaminergic LMP-inducing drug efficiently eradicated invasive glioma cells and secondary tumours within the brain. This unexpected fragility of the aggressive infiltrating cells to LMP provides new opportunities for clinical interventions, such as repositioning of an established antihistamine drug, to eradicate the inoperable, invasive, and chemo-resistant glioma cells from sustaining disease progression and recurrence.

**Keywords** antihistamine; glioma; LMP; MDGI; PUFA
**Subject Category** Cancer

## Introduction

Gliomas constitute approximately 30% of all primary nervous system tumours and 80% of all malignant brain tumours (Goodenberger & Jenkins, 2012). Glioblastoma is the most frequent, aggressive and lethal type of gliomas. Glioblastomas harbour a dense abnormal vasculature, display large hypoxic and necrotic areas and contain extensively proliferating tumour cells with the intrinsic ability to disseminate and colonize the brain far beyond the primary tumour mass. The current standard of care, comprising surgery, radio- and chemotherapy, provides only modest improvement in patient survival, and the prognosis remains dismal (Weller *et al*, 2017). This is due to (i) difficulty of complete surgical resection of the tumour, (ii) intratumoural heterogeneity and presence of multidrug-resistant cells and stem cell-like glioma cells responsible for the tumour maintenance and relapse, and (iii) the sheltering effect of the blood–brain-barrier (BBB), which efficiently prevents access of many systemic anti-cancer agents into the brain. Most probably, different approaches are required to eradicate the invasive cells and the cells that reside within the tumour bulk (Guishard *et al*, 2018). Thus, novel therapeutic approaches for glioblastomas are urgently needed (Reifenberger *et al*, 2017).

We have previously identified mammary-derived growth inhibitor (MDGI) as a glioma biomarker expressed in tumour cells and their associated vasculature (Hyvönen *et al*, 2014). MDGI, also known as heart-type fatty acid binding protein (H-FABP/*FABP3*), belongs to the family of fatty acid binding proteins (FABPs) that facilitate the intracellular transport of fatty acids (Glatz & van der Vusse, 1996). Both tumour-suppressive (Nevo *et al*, 2010) and tumour-promoting (Hashimoto *et al*, 2004; Sumantran *et al*, 2015) functions, depending on the cancer type, have been reported for MDGI. In glioma cells, MDGI has been found to mediate lipid droplet formation and fatty acid uptake (Bensaad *et al*, 2014) with the highest binding affinities to polyunsaturated fatty acids (PUFAs; Richieri *et al*, 2000).

1 Translational Cancer Medicine Research Program, Faculty of Medicine, University of Helsinki, Helsinki, Finland
2 Helsinki University Lipidomics Unit, Helsinki Institute of Life Science (HiLIFE) and Molecular and Integrative Biosciences Research Programme, University of Helsinki, Helsinki, Finland
3 Research Programs Unit, Genome-Scale Biology, University of Helsinki, Helsinki, Finland
4 Department of Microbiology, Tumour and Cell Biology (MTC), Karolinska Institutet, Stockholm, Sweden
5 Department of Neurology and Brain Tumour Center, University Hospital Zurich and University of Zurich, Zurich, Switzerland
6 Department of Oncology, Helsinki University Hospital, Helsinki, Finland
7 Department of Pathology, Haartman Institute, University of Helsinki and HUSLAB, Helsinki, Finland
8 Kuopio University Hospital, Kuopio, Finland
9 Laboratory Animal Centre, Helsinki Institute of Life Science (HiLIFE), University of Helsinki, Helsinki, Finland
*Corresponding author. Tel: +358 2 941 58100; Fax: +358 9 19125510; E-mail: Pirjo.laakkonen@helsinki.fi
[†]These authors contributed equally to this work

Here, we used cohorts of patients operated for primary glioma and patient-derived human spheroid cultures to examine the function of MDGI. Primary gliomas abundantly expressed MDGI, and high expression was associated with poor patient survival. MDGI was upregulated by hypoxia, and its overexpression enhanced the invasive growth of glioma cells both *in vitro* and *in vivo*. Surprisingly, MDGI silencing compromised spheroid growth and glioma cell survival via lysosomal membrane permeabilization (LMP). Lipid analyses of the lysosomal membranes showed that linoleic acid (18:2n-6), the major PUFA in the cell culture medium, was inefficiently incorporated into lysosomal phospholipids in MDGI-silenced cells. This led to biased molecular species composition of the phospholipids, in conjunction with reduced degree of lysosomal ceramide unsaturation. In addition, we show that glioma cells were more sensitive than normal cells to an LMP-inducing drug, the antihistamine clemastine, *in vitro,* and that clemastine treatment eradicated the invasive glioma cells *in vivo*. Our results suggest that MDGI expression is crucial for glioma cell viability and an important regulator of lysosomal integrity. The vulnerability of invasive glioma cells to lysosomal membrane destabilization opens new opportunities for LMP-inducing drugs as a promising treatment option.

# Results

## High MDGI expression correlates with poor glioma patient survival

Our previous results show that MDGI is expressed in a grade-dependent manner in human gliomas and its expression positively correlates with the histologic grade (Hyvönen *et al*, 2014). To study the potential correlation of MDGI expression with clinicopathological variables and patient survival, we performed immunohistochemistry for MDGI in human tumour microarrays (TMAs) consisting of lower WHO grade (grade II–III) gliomas and glioblastomas (grade IV; Fig 1A), and scored the staining intensity separately in tumour cells and tumour-associated endothelial cells. Approximately 50% of both grade II–III gliomas and glioblastomas expressed moderate to high levels of MDGI, accompanied with positive vascular staining for MDGI (Appendix Tables S1 and S2). Only 5% of all gliomas ($n$ = 6/122) showed no detectable MDGI expression. In the glioblastoma specimens, MDGI expression correlated with the presence of the CD117/C-Kit receptor in the perinecrotic tumour regions ($P$ = 0.006; Appendix Table S2). MDGI expression did not associate with the expression of EGFR ($P$ > 0.999), EGFRvIII ($P$ = 0.613), phosphorylated EGFR (Tyr-1173; $P$ > 0.999) or p53 ($P$ = 0.499; Appendix Table S2).

In lower grade gliomas (grade II–III), moderate to high MDGI levels significantly associated both with poor glioma-specific (HR = 1.85; 95% CI: 1.08–3.16; $P$ = 0.022; Appendix Fig S1A) and poor overall survival of patients (HR = 1.98; 95% CI: 1.19–3.28; $P$ = 0.007; Fig 1B) as compared to patients with negative or low tumour MDGI expression. Multivariate Cox hazards analysis showed that both MDGI expression and high tumour grade independently associated with unfavourable overall survival, increasing the risk of death by the factor of 2 (Table 1). In glioblastomas, tumour MDGI levels were not associated with overall survival (Appendix Fig S1B).

Next, we studied MDGI/*FABP3* expression in human glioma specimens and its association with survival using the GlioVis dataportal (http://gliovis.bioinfo.cnio.es). We analysed MDGI/*FABP3* expression in The Cancer Genome Atlas (TCGA) RNA seq datasets for glioblastoma (TCGA GBM) and for glioblastoma and low-grade glioma (TCGA GBMLGG). In the TCGA GBMLGG dataset, high MDGI mRNA expression associated with poor survival (Fig 1C), while no significant association between MDGI expression and patient survival was observed in the TCGA GBM dataset (Appendix Fig S1C). Thus, these results corroborate our immunohistochemistry results. When we analysed MDGI expression in the different histological glioma subclasses (grades II–IV), significantly more MDGI was expressed in glioblastomas compared to the lower grade gliomas (Appendix Fig S1D). When different glioblastoma subtypes were analysed, highest MDGI expression was observed in the mesenchymal subtype compared to the classical or pro-neural ones (Fig 1D). However, it did not reach the statistical significance. Moreover, the vast majority (94%) of MDGI-expressing glioblastomas displayed the non-G-CIMP phenotype (Appendix Fig S1E). In addition, in the lower grade gliomas, no significant difference in MDGI expression was observed between the IDH wt and mutant tumours (Appendix Fig S1F).

We then analysed MDGI expression using the Ivy Glioblastoma Atlas project (Ivy_GAP; http://glioblastoma.alleninstitute.org) RNA seq dataset, which maps gene expression across the anatomic structures and putative cancer stem cell clusters in glioblastomas. Interestingly, MDGI mRNA was expressed at significantly higher levels in the leading edge of the tumour and in infiltrative tumour cells compared to the microvascular proliferation, pseudopalisading cells or cells in the tumour mass (Fig 1E). In addition to the patient tissue biopsies, MDGI was expressed in all seven distinct patient-derived spheroid cultures containing stem cell-like glioma cells, whereas it was very low in 4 of 5 adherent cell lines studied (Fig 1F).

Our immunohistochemical results in clinical tumour samples revealed a correlation between MDGI expression and perinecrotic C-Kit, which is an indirect hypoxia marker in glioblastomas (Sihto *et al*, 2007). In patient-derived BT12 and BT13 spheroids, MDGI expression was high in conjunction with strong expression of hypoxia-inducible factor 1α (HIF-1α; Fig 1G). Addition of medium supplemented with serum led to drastic reduction (70–90%) in both MDGI and HIF-1α levels (Fig 1G) and shifted the growth from spheres to adherent monolayers. Moreover, MDGI expression was highly increased by hypoxia in adherent BT5, BT5R and U87MG cell lines (Fig 1H). No increase in MDGI expression in response to hypoxia was observed in BT12 and BT13 spheroid cultures, which already express high levels of MDGI and HIF-1α under normoxic conditions (Fig 1G and H). These results in glioma cells link MDGI induction to hypoxia.

## MDGI overexpression promotes glioma cell invasion

As MDGI expression was associated with glioma cell invasion and poor patient survival, we next studied MDGI function in glioma cell growth and invasion. We overexpressed MDGI as a GFP-fusion protein (MDGI-GFP) in U87MG cells since they express low endogenous levels of MDGI and form local, non-invasive tumours following intracranial injection in preclinical models (Lee *et al*, 2006; Xie *et al*, 2015). While MDGI overexpression did not affect cell proliferation (Fig EV1A), it significantly enhanced colony formation, suggesting an increased capacity for aggressive, anchorage-

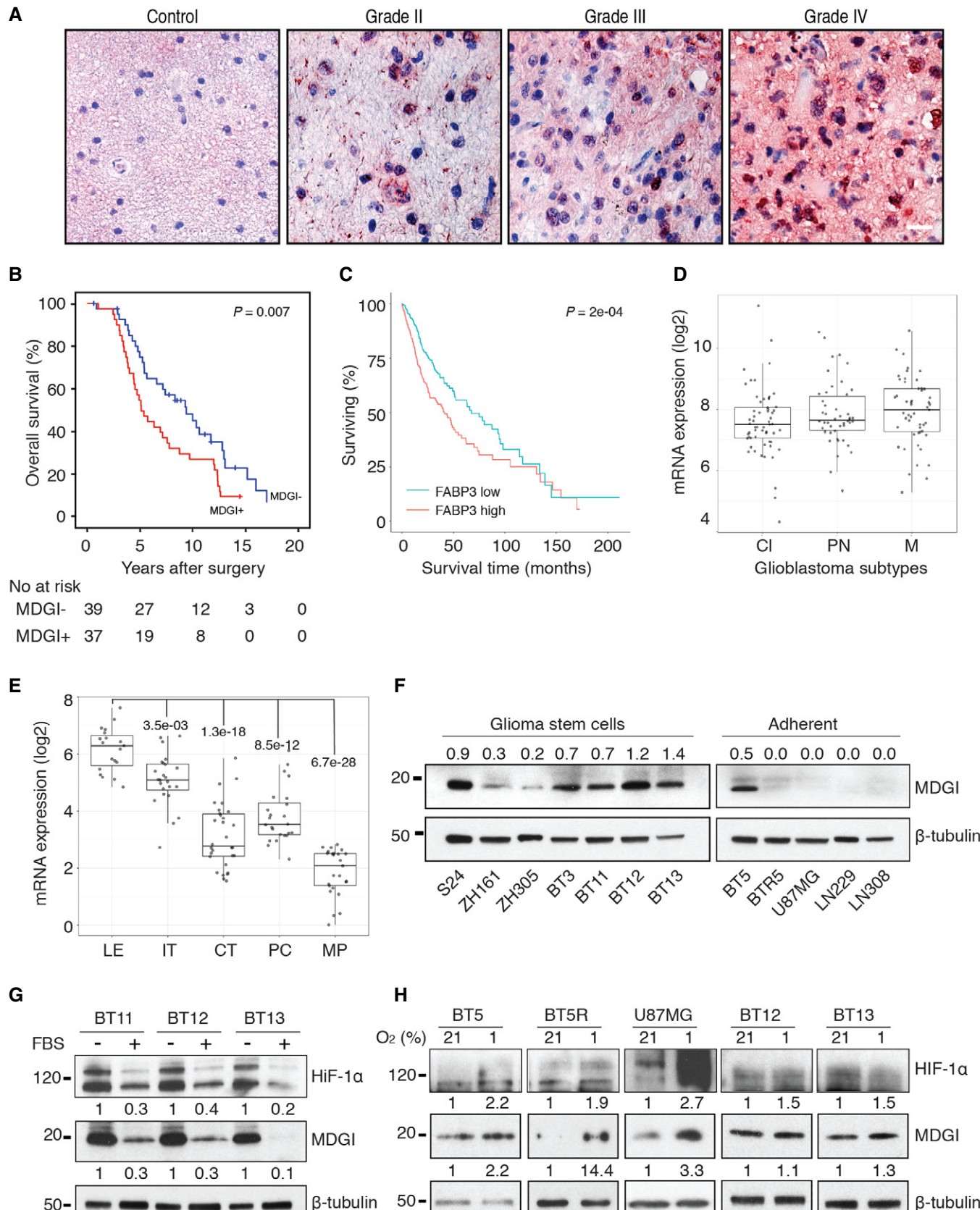

Figure 1.

**Figure 1. Association of MDGI protein expression with patient survival and regulation of MDGI expression by hypoxia.**

A  Glioma tumour microarrays (grade II–IV gliomas) were stained to visualize MDGI (red). Samples from the epileptic brain served as a negative control. Scale bar 100 μm.

B  Overall survival of grade II and III glioma patients ($n = 76$) was significantly better when none/low MDGI (blue line) was detected compared to patients with moderate/high MDGI (red line) expression ($P = 0.007$). The cumulative survival rates were estimated by using the Kaplan–Meier method and the statistical significance calculated by using the Mantel-Cox Log-rank test.

C  Analysis of the association of MDGI mRNA expression with glioma patient survival using the GlioVis data portal. In the TCGA GBMLGG dataset ($n = 667$), patients with low MDGI expression (green line; $n = 333$, events = 101, median = 67,5) show significantly better survival than patients with high MDGI (red line; $n = 334$, events = 138, median = 41.5) expression ($P = $ 2e-04). HR = 0.62 (0.48–0.8). The cumulative survival rates were estimated by using the Kaplan–Meier method and the statistical significance calculated by using the Mantel-Cox Log-rank test.

D  In glioblastoma samples, highest MDGI expression was observed in the mesenchymal subtype (M) compared to the classical (Cl) or pro-neural (PN) ones. However, it did not reach the statistical significance. GlioVis data portal, TCGA GBM dataset, $n = 156$. Pairwise $t$-test with corrections for multiple testing, $P$-values with Bonferroni correction. In the boxplot, line represents the median, box limits represent the lower and upper quartiles, and whiskers represent outlier values.

E  MDGI expression was high in the leading edge of the tumour (LE) and in infiltrating tumour cells (IT) in the Ivy_Gap dataset ($n = 122$) where the gene expression profile in different anatomical structures of glioblastomas was analysed. LE, leading edge; IT, infiltrating tumour cells; CT, cellular tumour (cells within the tumour mass); PC, pseudopalisading cells; MP, microvascular proliferation. Numbers in the graph depict the $P$-values: pairwise $t$-test with corrections for multiple testing, $P$-values with Bonferroni correction. In the boxplot line represents the median, box limits represent the lower and upper quartiles, and whiskers represent outlier values.

F  Western blot analysis shows MDGI expression in human glioma cells including patient-derived spheroids (S24, ZH161, ZH305, BT3, BT11, BT12, BT13), patient-derived adherent cells (BT5 and BT5R) and long-term cell lines (U87MG, LN229, LN308). Numbers above the lanes show the relative expression of MDGI compared to the levels of β-tubulin that served as loading control in each cell line. Representative Western blot image of three separate experiments is shown.

G  High MDGI and HIF-1α expression in the patient-derived glioblastoma spheroids decreased significantly when cells were cultured in medium containing 10% of serum (FBS) for 7 days. Numbers show the relative expression of MDGI and HIF-1α compared to the levels of β-tubulin. Expression in the serum-free medium was set as 1. Representative Western blot image of two separate experiments is shown.

H  MDGI expression in adherent glioma cell lines and in glioma cell spheroids after 24-h culture under hypoxia (1% $O_2$) or normoxia (21% $O_2$). Numbers show the relative MDGI expression compared to the levels of β-tubulin. Expression under normoxia was set as 1. Representative Western blot image of three separate experiments is shown.

Source data are available online for this figure.

**Table 1.  Cox multivariate analysis of association of MDGI expression and tumour grade on glioma patient survival.**

| Covariate | β (SE) | HR | P |
|---|---|---|---|
| MDGI expression | | | |
| Positive vs. negative | 0.56 (0.264) | 1.76 (1.05–2.94) | 0.033 |
| Tumour grade | | | |
| III vs. II | 0.76 (0.264) | 2.14 (1.28–3.60) | 0.004 |

β = regression coefficient of hazard function; HR = hazard ratio.

independent growth of the MDGI-overexpressing cells (Fig EV1B and C). In addition, MDGI-overexpressing spheroids grew more invasively in an *ex vivo* brain slice model than the control cells (Fig EV1D and E). Moreover, the intracranial U87MG-MDGI-GFP xenografts grew invasively (Fig 2A and B), formed secondary tumours (diameter > 300 μm) in the brain (Fig 2C, D and G) and displayed vascular co-option (angiotropic tumours with diameter < 300 μm, Fig 2E, F and H) unlike the control GFP-expressing U87MG-derived xenografts that only formed well-delineated masses. Next, we overexpressed MDGI in the LN229 glioblastoma cells that in addition to formation of the primary tumour mass invade into the brain parenchyma and form secondary vasculature-associated angiotropic tumours. Also, in this model, high MDGI expression significantly promoted the invasion and formation of angiotropic tumours (Fig 2I–P) consistent with the results obtained with the U87MG-MDGI-GFP xenograft model.

## MDGI silencing reduces glioblastoma cell viability

To investigate the functions of the endogenous MDGI in glioblastoma cells, we silenced MDGI in patient-derived BT12 and BT13 cells using two different shRNAs (shMDGI1 and shMDGI2). The shMDGI1 resulted in undetectable MDGI levels, while shMDGI2 showed approximately 60–70% knockdown efficiency (Fig EV2A and B). MDGI silencing caused a dramatic change in the cell morphology abrogating the formation of the large multicellular spheroids (Fig 3A). Moreover, the self-renewal ability in methylcellulose (Fig 3B and C) and the anchorage-independent growth in soft agar (Fig EV2C) of the MDGI-silenced cells were severely compromised. Surprisingly, MDGI silencing inhibited proliferation of both BT12 and BT13 cells (Figs 3D and EV2D) and dramatically reduced their viability (Fig EV2E). When the MDGI-silenced cells were injected intracranially into immunocompromised mice, no tumour formation was observed, while the control shRNA (Scr)-infected cells formed invasive tumours (Figs 3E and F, and Appendix Fig S2A–C). These results demonstrate a dose-dependent effect of MDGI silencing on glioblastoma cell growth and viability.

MDGI has been previously shown to affect EGFR trafficking (Nevo *et al*, 2009), inhibition of which can induce glioma cell apoptosis (Ghildiyal *et al*, 2013; Kaluzova *et al*, 2015). Thus, we investigated the EGFR expression and pathway activity in glioma cells. Only the patient-derived glioblastoma BT11, BT12 and BT13 spheroids expressed high levels of EGFR (Appendix Fig S2D). As expected, BT12 and BT13 cells were highly sensitive to the commonly used EGFR inhibitor, gefitinib (Wakeling *et al*, 2002). While 0.4 μM of gefitinib was able to kill up to 50% of the BT12 and BT13 cells, 23.5-fold higher concentration (9.4 μM) was required to kill 50% of the ZH305 cells that expressed low levels of EGFR (Appendix Fig S2E). EGFR expression was significantly reduced in response to MDGI silencing in BT12 and BT13 cells both at the mRNA and protein levels (Appendix Fig S2F and G). However, MDGI silencing also induced death of the ZH305 spheroids expressing low levels of EGFR, suggesting that MDGI silencing-triggered cell death was independent of EGFR expression (Fig EV2F and G).

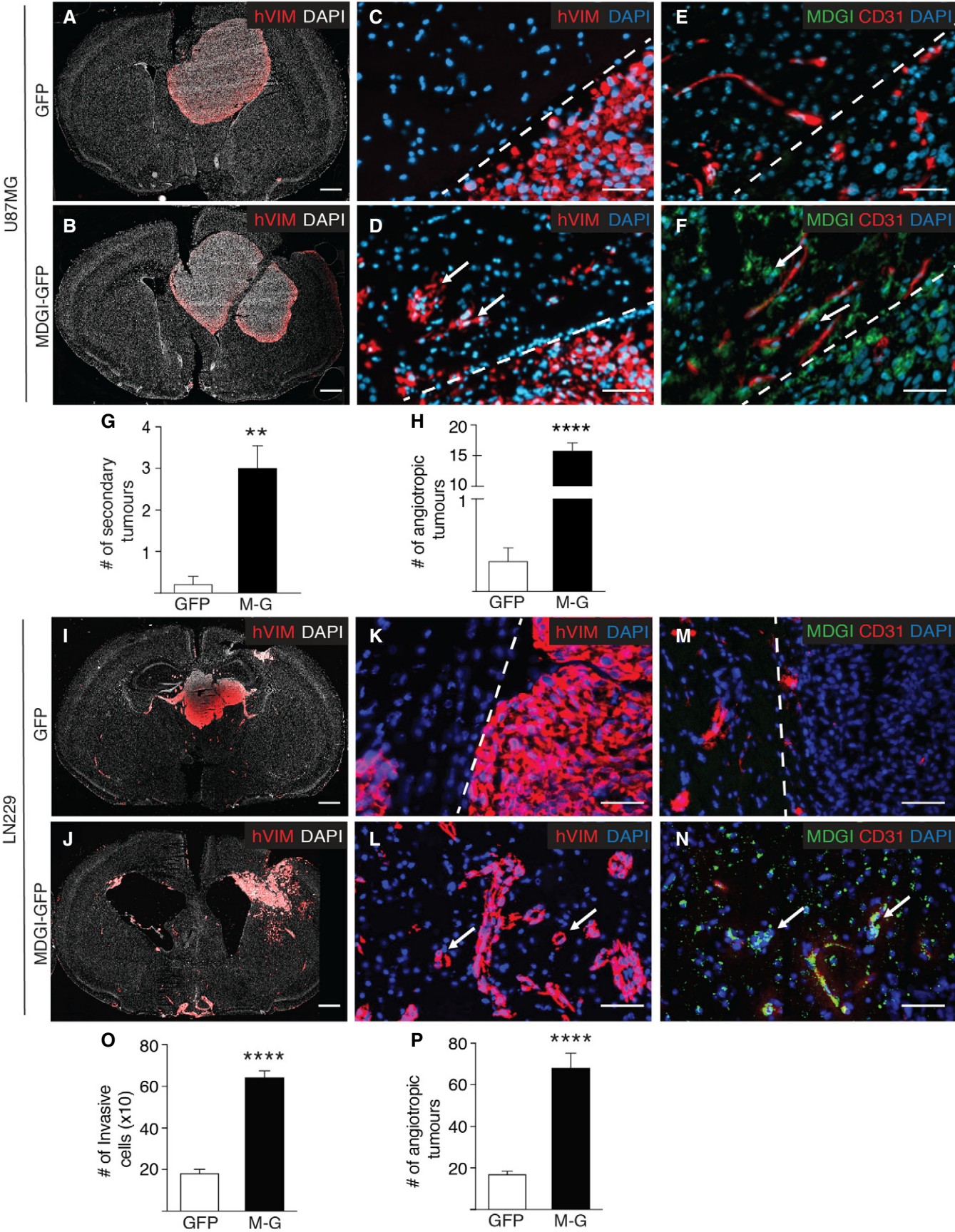

**Figure 2.**

**Figure 2. MDGI expression promotes aggressive and invasive growth of glioma cells.**

A, B    Representative whole coronal section micrographs of murine brain injected with GFP (A) or MDGI-GFP (B) expressing U87MG glioma cells. Xenografted human cells were visualized by using anti-human vimentin (hVim) antibodies (red). Nuclei were visualized by using DAPI (white). Scale bar: 1 mm.

C–F    Micrographs of consecutive (separating distance: 9 μm) brain sections stained for human glioma cells using antibodies specific for human vimentin (hVim red in C and D), MDGI (green in E and F) and CD31 (red in E and F). Dashed lines separate the primary tumour mass and the normal brain. Nuclei were visualized by using DAPI (blue). Arrows in (D) point to the angiotropic tumours. Arrows in (F) point to the MDGI-expressing angiotropic tumour cells. Scale bar: 50 μm.

G    Quantification of the number of U87MG secondary tumours (diameter > 300 μm) detected in the whole brain (GFP: n = 5, MDGI-GFP: n = 9). Data are represented as mean ± SEM. **$P < 0.01$, two-tailed, nonparametric Mann–Whitney's U-test.

H    Quantification of the number of invasive U87MG cells that grew next to brain blood vessels (angiotropic co-opting tumour cells, diameter < 300 μm; GFP: n = 5, MDGI-GFP: n = 9). Data are represented as mean ± SEM. ****$P < 0.0001$, two-tailed, nonparametric Mann–Whitney's U-test.

I, J    Representative micrographs of whole murine brain coronal sections injected with GFP (I) or MDGI-GFP (J) expressing LN229 glioma cells. Xenografted human cells were visualized by using anti-human vimentin (hVim) antibodies (red). Nuclei were visualized by using DAPI (white). Scale bar: 1 mm.

K–N    Micrographs of consecutive (separating distance: 9 μm) brain sections stained for human glioma cells using antibodies specific for human vimentin (hVim red in K and L), MDGI (green in M and N) and CD31 (red in M and N). Dashed lines separate the primary tumour mass and the normal brain. Nuclei were visualized by using DAPI (blue). Arrows in (L) point to the angiotropic tumours. Arrows in (N) point to the MDGI-expressing angiotropic tumour cells. Scale bar: 50 μm.

O    Quantification of the number of invasive LN229 cells detected in the brain (GFP: n = 10, MDGI-GFP: n = 10). Data are represented as mean ± SEM. ****$P < 0.0001$, two-tailed, nonparametric Mann–Whitney's U-test.

P    Quantification of the number of invasive LN229 cells that grew next to brain blood vessels (angiotropic co-opting tumour cells, diameter < 300 μm) detected in the brain (GFP: n = 10, MDGI-GFP: n = 10). Data are represented as mean ± SEM. ****$P < 0.0001$, two-tailed, nonparametric Mann–Whitney's U-test.

Source data are available online for this figure.

## MDGI silencing induces caspase-independent death of glioma cells

To further examine the mechanism underlying the reduced viability of MDGI-silenced cells, we assessed the Annexin V binding to the cell surface-exposed phosphatidylserines as a marker of cell death (Koopman et al, 1994). MDGI silencing significantly increased the binding of Annexin V to MDGI-silenced BT12 cells compared to the control cells, indicating increased apoptosis (Fig 4A and B). To study the intracellular pathways that could contribute to this, we analysed the expression of apoptosis-associated proteins at various time points after lentiviral MDGI silencing. Levels of the phosphorylated and total p53 and the pro-apoptotic protein BAD first remained stable but eventually decreased 5–6 days after transduction (Appendix Fig S3A). Surprisingly, the amounts of total caspase-3 and its cleaved, activated form also decreased 5–6 days after silencing (Appendix Fig S3A). We also studied the expression of selected proteins of the key signalling pathways at different time points after MDGI silencing using Western blot. In MDGI-silenced cells, p27 levels were elevated and the Erk1/2 phosphorylation was significantly induced, while the phosphorylated AKT was decreased already at day three after silencing (Appendix Fig S3B). In addition, in an antibody array of apoptosis-associated proteins, levels of the anti-apoptotic proteins survivin and X-linked inhibitor of apoptosis protein (XIAP) decreased in MDGI-silenced cells. We also observed a decrease in the active caspase-3 level. However, the fold-change (0.61) exceeded the threshold, and therefore, it was considered as unchanged. On the other hand, levels of the HTRA serine protease were elevated, while levels of pro-apoptotic proteins BAD and BAX or cytochrome C were unchanged by MDGI silencing (fold-change less than 0.6 or more than 1.5; Appendix Fig S3C and D). Furthermore, MDGI silencing-induced cell death could not be rescued by siRNA silencing of the pro-apoptotic protein BAD (Appendix Fig S3E and F). This suggests that apoptosis in the MDGI-silenced cells was not mediated by caspase activation.

## MDGI silencing induces lysosomal membrane permeabilization (LMP)

Since apoptosis in MDGI-silenced cells was not mediated by caspase activation, we studied the effects of MDGI silencing on LMP. This alternative cell death pathway leads to release of lysosomal hydrolases into the cytosol, and depending on the extent of the release can ultimately lead to lysosomal cell death with necrotic or apoptotic features (Aits & Jaattela, 2013). LMP induction can be easily visualized as a change in galectin-1 localization from a diffuse cytoplasmic to a punctate staining pattern (Aits et al, 2015). As a control, we first treated the glioma spheroids with an LMP-inducing agent, L-leucine O-methyl (LLOMe; Uchimoto et al, 1999), which caused increased formation of galectin-1 (LGALS1)-positive puncta (Fig 4C and D). Similarly, we detected a significant increase in the number of LGALS1-positive punctate staining in the MDGI-silenced patient-derived BT12 and BT13 cells (Fig 4E–I). In addition, a significant fraction of the MDGI-silenced cells was already dead at the time of analysis, as revealed by the quantification of the number of fragmented nuclei devoid of LGALS1-staining (Fig 4J). LGALS1-puncta co-localized with the lysosomal marker protein, lysosome-associated membrane protein 2 (LAMP2), verifying the lysosomal localization of LGALS1 (Fig EV3A). Moreover, the activity of lysosomal cathepsin B was significantly increased in the cytoplasm of the MDGI-silenced cells (Fig 4K). Interestingly, when we cultured the BT12 and BT13 cells with the pan-cathepsin inhibitor (K777), survival of MDGI-silenced cells was partially rescued (Fig 4L). Only 11% of the MDGI-silenced cells were alive at day 5 after shRNA transduction compared to the control cells, while 41% of the MDGI-silenced cells were alive when cultured in the presence of K777 compared to the control cells (Fig 4L).

## MDGI silencing changed lysosomal membrane lipid composition

To investigate whether MDGI silencing would affect the lysosomal membrane lipid composition due to its fatty acid binding and transport function, we enriched lysosomes by density gradient ultracentrifugation (Fig EV4B) and analysed the extracted lysosomal lipids by mass spectrometry. The lipid classes with most evident compositional differences between the MDGI-silenced and control cells were phosphatidylcholine (PC), phosphatidylethanolamine (PE) and ceramide. PC and PE are the bulk lipids of the membrane that together comprise about 70% of the lysosomal membrane phospholipids (Kobayashi et al, 2002). Lysosomes of the MDGI-silenced cells

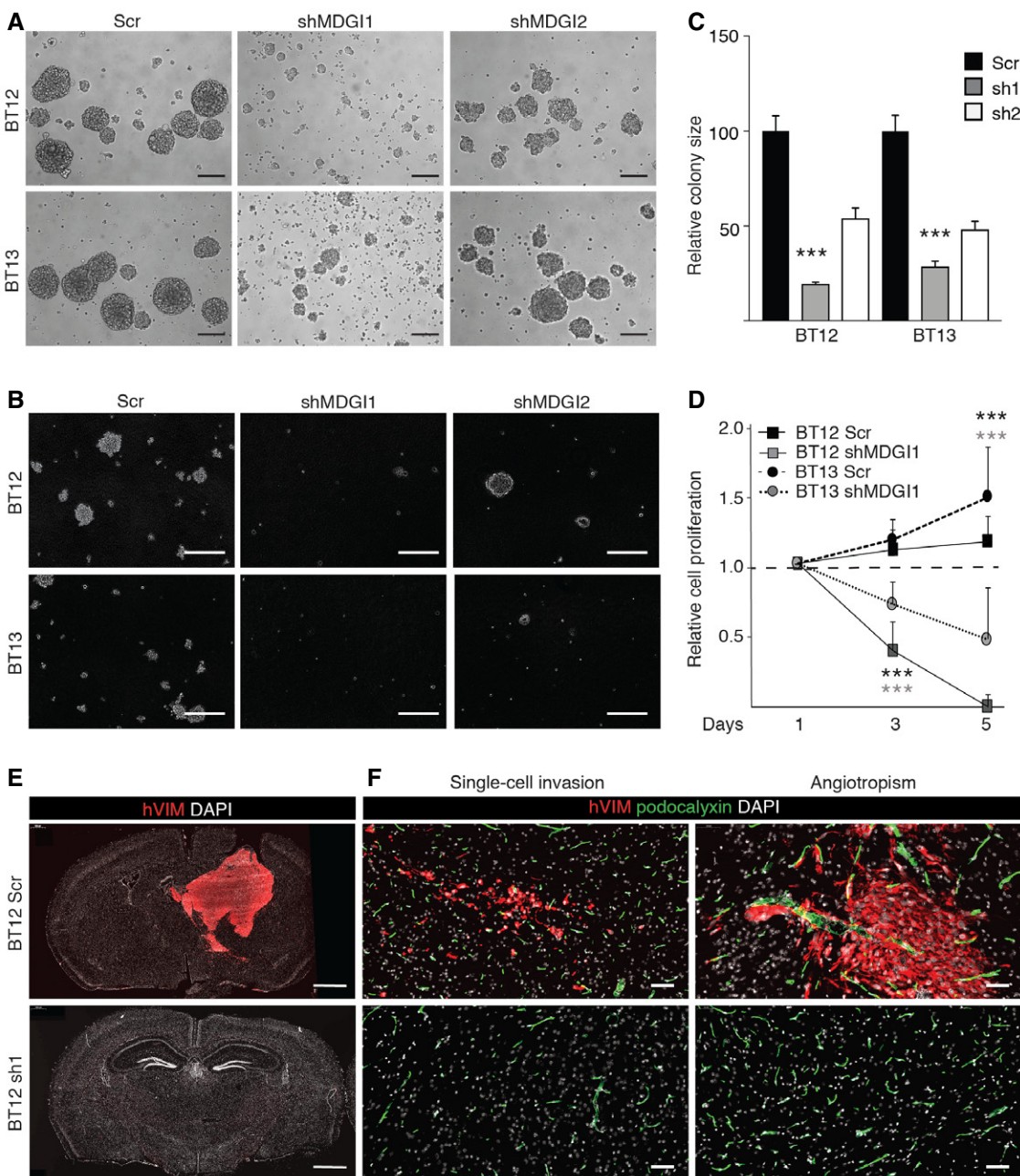

**Figure 3.   MDGI silencing reduces glioma cell growth and viability both *in vitro* and in *in vivo*.**

A    Transmitted light micrographs of MDGI-silenced (shMDGI1 and shMDGI2) and control (Scr) patient-derived BT12 and BT13 spheroids 11 days after gene silencing. Representative images of at least three independent experiments are shown. Scale bar: 200 μm.

B    Methylcellulose assay demonstrating the ability for self-renewal of control (Scr) and MDGI-silenced (shMDGI1 and shMDGI2) patient-derived BT12 and BT13 cells after 3 weeks of culture. Representative images of at least three independent experiments are shown. Scale bar: 200 μm.

C    Quantification of the colony size in methylcellulose of MDGI-silenced (sh1 and sh2) cells relative to the controls (Scr) that is set as 100 ($n = 3$). Data are represented as mean $\pm$ SD. ***$P < 0.001$, two-tailed, nonparametric Mann–Whitney's *U*-test.

D    MTT-proliferation assay of control (Scr) and MDGI-silenced (shMDGI1) BT12 and BT13 cells. The values of each cell line were normalized to the values at day 1 marked with a dashed line ($n = 3$). Data are represented as mean $\pm$ SD. ***$P < 0.001$, two-tailed, nonparametric Mann–Whitney's *U*-test.

E, F   Representative whole coronal section micrographs of murine brain injected intracranially with control (BT12 Scr, $n = 5$) and MDGI-silenced (BT12 sh1, $n = 5$) BT12 cells. After 4 weeks, animals were sacrificed, and the brains were excised and prepared for immunohistochemistry. Control (Scr) cells grew very similarly to the parental BT12 cells forming a tumour mass with invasive tumour cells (single-cell invasion) that eventually co-opted existing blood vessels (angiotropism) and formed secondary tumours (upper panels). No tumour growth was observed when MDGI-silenced cells were implanted (lower panels). Glioma cells were visualized by using anti-human vimentin (hVim red) and blood vessels using anti-podocalyxin antibodies (green). Nuclei were visualized by using DAPI (white). Scale bar: 1 mm (E) and 50 μm (F).

Source data are available online for this figure.

contained decreased proportions of di-unsaturated PC and PE species compared to the lysosomes of the control cells (Fig 5A), especially the 36:2 was diminished due to the silencing (Fig 5B). At the same time, the proportions of monounsaturated PC and PE species (34:1 and 36:1) increased and those of saturated PC species (30:0 and 32:0) decreased compared to the lysosomes of the control cells (Fig 5B). While no differences were observed in the total proportions of the more highly unsaturated PC and PE species (double bond content 3–7) in response to MDGI silencing (Fig 5A), certain polyunsaturated species with 3 or 4 double bonds were increased (PE 38:3, 40:3, 36:4, 38:4) in conjunction with decrease in species with 5–7 double bonds (PE 40:5, 40:7; Fig 5B).

Ceramides (Cer), regarded as important determinants of membrane permeability, have only one N-linked acyl chain, which commonly is either saturated or monounsaturated. Lysosomes of the MDGI-silenced cells showed a marked compositional change of ceramide species (Fig 5C). In the control cell lysosomes, Cer 24:1 dominated by threefold over Cer 24:0, while in the lysosomes of the MDGI-silenced cells these two Cer species were found in equal amounts (Fig 5C and D). The consequent impairment of the elastic properties of membrane bilayer (Bruno *et al*, 2007; Pan *et al*, 2009) probably contributed to the found leakage of lysosomal membranes.

### Clemastine evokes glioma cell death

Inspired by MDGI silencing-triggered cell death via LMP, we searched for an LMP-inducing drug that could be safely used in preclinical studies. A recent report by Ellegaard *et al* (2016) identified cationic amphiphilic (CAD) antihistamines as drugs able to provoke LMP. Therefore, we chose clemastine (Tavegil™), a first-generation histamine H1 blocking antihistamine CAD, as the BBB-permeable drug for our experiments. The patient-derived BT12, BT13 and ZH305 glioblastoma cells, as well as various normal cells, were treated with increasing concentrations of clemastine (1–5 μM). The highest concentration killed all the cells *in vitro* already by day 3 (Fig 6A and B). About 90% cell death was observed with 2 μM clemastine concentration, while 1 μM clemastine concentration killed 50% of BT12 and BT13 cells and 64% of ZH305 cells at day 4 (Fig 6A). No significant cell death was observed when normal human endothelial cells (HuAR2T), normal human astrocytes (NHA), embryonic kidney (HEK293T; Fig 6B) or murine brain endothelial (Fig EV3C) cells were treated at 1–2 μM of clemastine, suggesting a therapeutic window for clemastine treatment in gliomas. In accordance, already 1 μM of clemastine induced punctate localization of the galectin-1 in BT12, BT13 and ZH305 cells (Fig 6C), whereas no galectin-1 re-localization was observed in HuAR2T, NHA or HEK293T cells (Fig 6D). Galectin-1 relocation into the lysosomes cells was confirmed by co-localization with the LAMP2 (Fig EV3D). Clemastine treatment had no effect on MDGI levels in glioblastoma cells *in vitro* (Appendix Fig S4A), suggesting that the clemastine effect is not upstream of MDGI.

The preclinical evaluation of clemastine was performed in patient-derived xenografts orthotopically implanted in immunocompromised mice. We chose three different patient-derived glioblastoma models with different intracranial growth patterns: BT13 cells grow as an angiogenic non-invasive tumour, ZH305 cells grow highly diffusively without formation of tumour bulk, and BT12 cells show a mixed phenotype with tumour bulk and invasive

growth. After 15 days of tumour growth, we started intraperitoneal injections of clemastine at a dose of 100 mg/kg on the first day followed by 50 mg/kg daily injections for 12 additional days. The vehicle (saline solution) was administered under the same modalities to the control cohort. As illustrated in Fig 7A, clemastine provoked a profound alteration in the growth pattern of BT12-, BT12 Scr- and ZH305-derived xenografts. No difference was observed in the BT13-derived xenografts that form bulky angiogenic tumours without invasive tumour cells (Fig 7A). The MDGI-silenced BT12 cells did not form tumours *in vivo* (Figs 3E, and EV4A and B). The number and distance of invasive cells that had escaped the primary tumour and disseminated into the brain were dramatically reduced (Fig 7B and C) to the extent that in the ZH305 model, no tumour cells were detected (Figs 7A and EV4C). In addition, clemastine treatment significantly reduced the number of secondary tumours in BT12 and BT12 Scr xenografts (Fig 7D). In addition, clemastine treatment led to a nearly complete loss of co-opting angiotropic tumour cells (Fig 7E). To further evaluate the anti-tumour effect of clemastine, we analysed the number of apoptotic cells by terminal deoxynucleotidyl transferase dUTP nick end labelling (TUNEL) in control and clemastine-treated BT12-xenograft bearing animals. Whole-brain quantification revealed that the leading, migratory edge of the tumour was more susceptible to the cytotoxic effect of clemastine (Fig 7F and G) than the cells inside the tumour mass. As a consequence, clemastine-treated animals showed significantly prolonged survival ($P = 0.044$) compared to the controls (Fig 7H). To study whether clemastine affected galectin-1 localization as a sign of LMP induction also *in vivo*, we analysed the BT12-derived control and clemastine-treated xenografts by using immunohistological and Western blot analyses. Clemastine induced re-localization of galectin-1 into a more punctate pattern (Fig EV4D and E) but did not affect the galectin-1 expression levels (Fig EV4F). No cytotoxic effect of clemastine apart from the transient drowsiness of the animals following drug injections was observed during the treatment. We weighted the animals during the treatment and post-mortem measured selected organs' weight. The vehicle-treated animal lost weight at the end of the treatment probably due to the tumour burden (Appendix Fig S4B), but no significant difference in the organ:body weight ratios was observed between the vehicle- and clemastine-treated animals (Appendix Fig S4C–F).

## Discussion

There is an unmet need for novel therapeutic strategies to treat gliomas, making the characterization of proteins involved in disease progression highly imperative. Here, we examined the expression of a potential glioma biomarker, MDGI (Hyvönen *et al*, 2014), in WHO grade II–III glioma and glioblastoma (grade IV) specimens, publicly available datasets as well as in patient-derived and commercially available glioblastoma cell lines. We show that high MDGI expression associated with unfavourable patient survival and was crucial for glioblastoma cell viability. We further describe a novel function for MDGI in the maintenance of lysosomal membrane integrity and show the exceptional vulnerability of invasive glioma cells to destabilization of lysosomes. In addition, we show that antihistaminergic drug clemastine eradicated invasive glioblastoma cells *in vivo* via

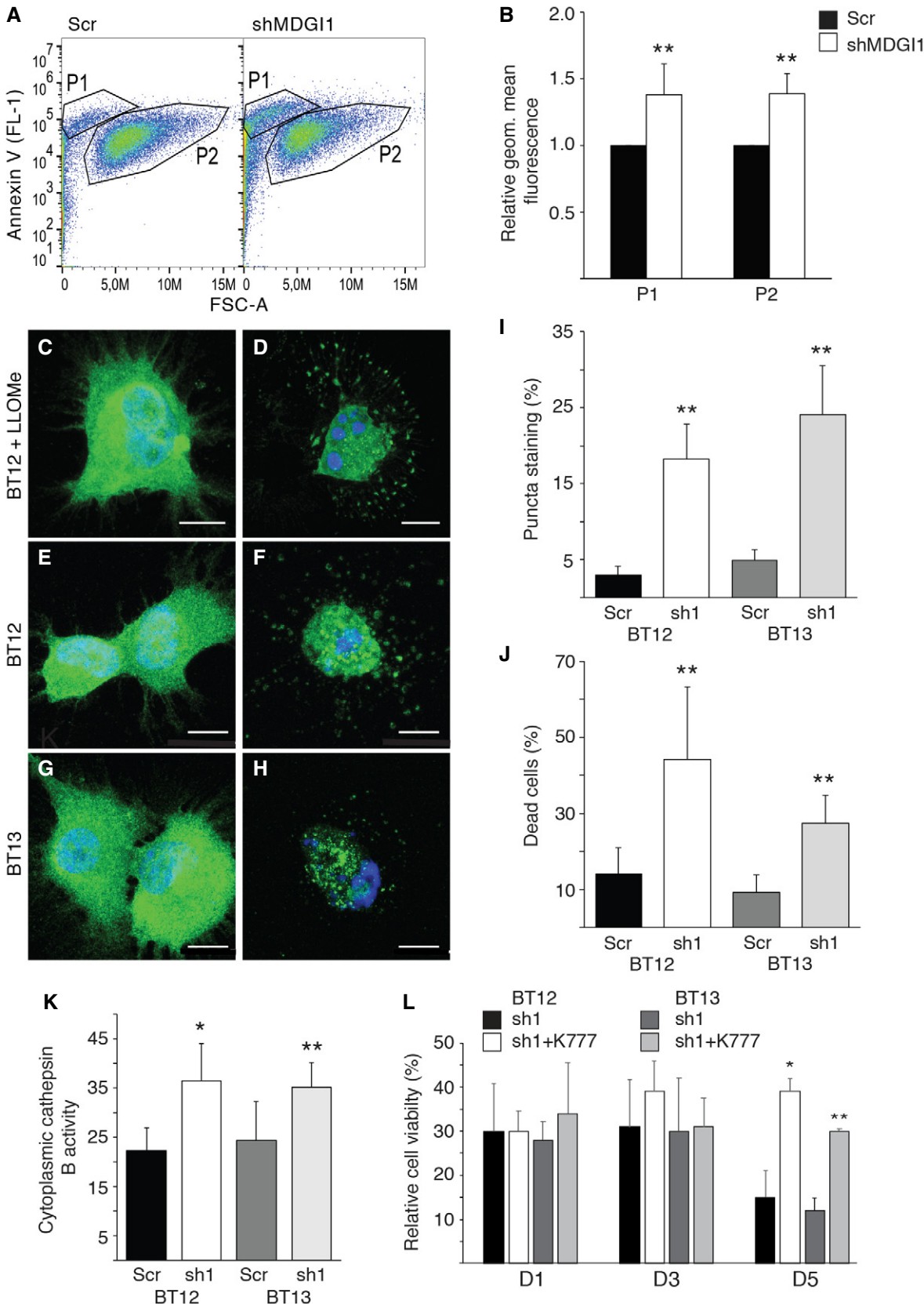

**Figure 4.**

◄

**Figure 4.   MDGI silencing induces cell death via lysosomal membrane permeabilization (LMP).**

A    Binding of Annexin V Alexa Fluor 488 to control (Scr) and MDGI-silenced (shMDGI1) BT12 cells was measured by flow cytometric analyses 5 days after silencing. The analysis was performed by gating the cells into two populations: P1 (small, granular) and P2 (normal-sized cells) (30,000 analysed events/cell line, n = 3).

B    Geometric mean fluorescence (FL-1) relative to the controls (Scr) that was set as 1 in P1 and P2 in MDGI-silenced (shMDGI1) cells. Experiment was repeated three times (n = 2). Data are represented as mean ± SD. **P < 0.01, two-tailed, nonparametric Mann–Whitney's U-test.

C, D   Representative micrographs of non-treated control BT12 cells (C) and cells treated with 2 mM LLOMe for 4 h (D) before staining with the anti-galectin-1 antibodies (LGALS1; green). Nuclei were visualized by using DAPI (blue). Scale bar: 10 μm.

E, F   Representative micrographs of LGALS1-stained (green) BT12 control (E) and MDGI-silenced cells (F) 6 days after silencing. Nuclei were visualized by using DAPI (blue). Scale bar: 10 μm.

G, H   Representative micrographs of LGALS1-stained (green) BT13 control (G) and MDGI-silenced cells (H) 6 days after silencing. Nuclei were visualized by using DAPI (blue). Scale bar: 10 μm.

I    Percentage of LGALS1-positive puncta staining in the control (Scr) and MDGI-silenced (sh1) cells. In total, 0.4–1.2 × 10⁴ LGALS1-stained cells were analysed from 50-mm² coverslip areas (n = 6). Data are represented as mean ± SD. **P < 0.01, two-tailed, nonparametric Mann–Whitney's U-test.

J    Quantification of the percentage of dead cells 6 days after MDGI silencing and LGALS1 staining (n = 100) in control (Scr) and MDGI-silenced (sh1) BT12 and BT13 cells. Data are represented as mean ± SD. **P < 0.01, two-tailed, nonparametric Mann–Whitney's U-test.

K    Cytoplasmic cathepsin B activity in the control (Scr) and MDGI-silenced (sh1) BT12 and BT13 cells (n = 4). Data are represented as mean ± SD. *P < 0.05; **P < 0.01, two-tailed, nonparametric Mann–Whitney's U-test.

L    Number of live cells in MDGI-silenced BT12 and BT13 cells incubated without (sh1) or with (sh1 + K777) the pan-cathepsin inhibitor (K777) after 1, 3 and 5 days of incubation. Results are presented as pooled values of 3 independent experiments (total n = 9). Data show the percentage of live cells compared to the control (Scr) cells set as 100%. Data are represented as mean ± SD. *P < 0.05; **P < 0.01, two-tailed, nonparametric Mann–Whitney's U-test.

Source data are available online for this figure.

induction of lysosomal membrane permeabilization and prolonged survival of glioblastoma-bearing mice.

Although MDGI expression has been linked to tumour-suppressive properties in breast cancer (Nevo *et al*, 2010), in gastric carcinomas MDGI associated with poor patient survival (Hashimoto *et al*, 2004) and in melanomas, its expression was upregulated during disease progression (Sumantran *et al*, 2015). Here, we show that MDGI was frequently expressed in human gliomas, and high MDGI expression significantly correlated with poor survival. These results suggest that the effect of MDGI on tumorigenesis may be tissue-dependent and cancer type-dependent. We detected high MDGI expression also in various patient-derived, cancer stem cell-enriched spheroids but not in adherent cells under normoxia. However, hypoxia stimulated MDGI expression in adherent cells. This is in agreement with previous data that showed upregulation of MDGI expression during hypoxia (Bensaad *et al*, 2014). MDGI overexpression in glioma cells significantly promoted their anchorage-independent growth and invasion both *in vitro* and *in vivo*, suggesting a functional role for MDGI in the invasive growth. Consistent with this, MDGI/*FABP3* expression was highest in the mesenchymal subtype of glioblastomas (GlioVis) as well as in the leading edge and infiltrating tumour cells based on tumour structure-specific mRNA expression dataset of glioblastomas (Ivy_GAP).

Interestingly, our results demonstrate that MDGI expression is crucial for glioma cell survival, and MDGI depletion induces lysosomal membrane permeabilization (LMP). LMP is an intracellular cell death pathway that can be either caspase-independent or caspase-dependent and it can occur either up- or downstream of mitochondrial membrane permeabilization depending on the cell type and/or LMP-inducer (Boya *et al*, 2003; Aits & Jaattela, 2013; Huai *et al*, 2013; Shen *et al*, 2013). As a consequence, LMP results in irreversible leakage of lysosomal proteolytic enzymes to the cytoplasm, where they digest vital proteins and intracellular organelles (Kirkegaard & Jaattela, 2009). Considering the function of MDGI as a fatty acid binding protein, we hypothesized that LMP may be induced by changes in lysosomal membrane lipid composition. Indeed, MDGI has been reported to

bind efficiently the polyunsaturated fatty acids (PUFAs; Richieri *et al*, 2000), and in the MDGI knockout mice, incorporation of PUFAs into phospholipid membranes was reduced (Murphy *et al*, 2005). Our lipid analyses show that MDGI silencing impaired trafficking of polyunsaturated fatty acids (FA) into cells resulting in significant alterations in the lipid composition of lysosomal membranes. Among the FAs preferred by MDGI is linoleic acid 18:2n-6, which cannot be synthesized *de novo* by the cells but has to be received from external sources. In fact, 18:2n-6 was the main FA component in cell culture media used in this study suggesting that MDGI silencing impaired the uptake and incorporation of the 18:2 FA chain into the lysosomal phospholipids. Mammalian cells actively regulate their membrane fluidity, and when the supply of exogenous PUFAs diminishes, the endogenous production of monounsaturated FAs (MUFAs) from the saturated FA (SFA) precursors by desaturases is activated (Ntambi, 1999). In line with this, the lysosomal membranes of MDGI-silenced cells contained larger proportions of MUFAs and fewer SFAs than the control lysosomes, which can be seen as fluidity compensation among these membrane bulk phospholipids. However, it should be recalled that MUFAs cannot replace the essential PUFAs in their specific biological functions, and it is likely that this compositional bias caused by MDGI silencing hampered lysosomal membrane integrity, dynamics and vesicle secretion, and changed the precursor pool of signalling molecules (Erazo-Oliveras *et al*, 2018).

Membrane ceramide content has been considered as an important factor defining membrane permeability, albeit the mechanistic explanations vary from pore induction to gradually increasing defects in lipid packing (Artetxe *et al*, 2017). The lysosomes of the MDGI-silenced cells showed no marked change in the total ceramide content or their acyl chain length distribution, commonly attributed to cell fate decisions and induction of apoptosis (Grosch *et al*, 2012). Our data, however, identified ceramides as potential mediators of LMP, since the degree of unsaturation of the lysosomal ceramides (ratio of Cer 24:1 to Cer 24:0) markedly decreased upon MDGI silencing. These observed

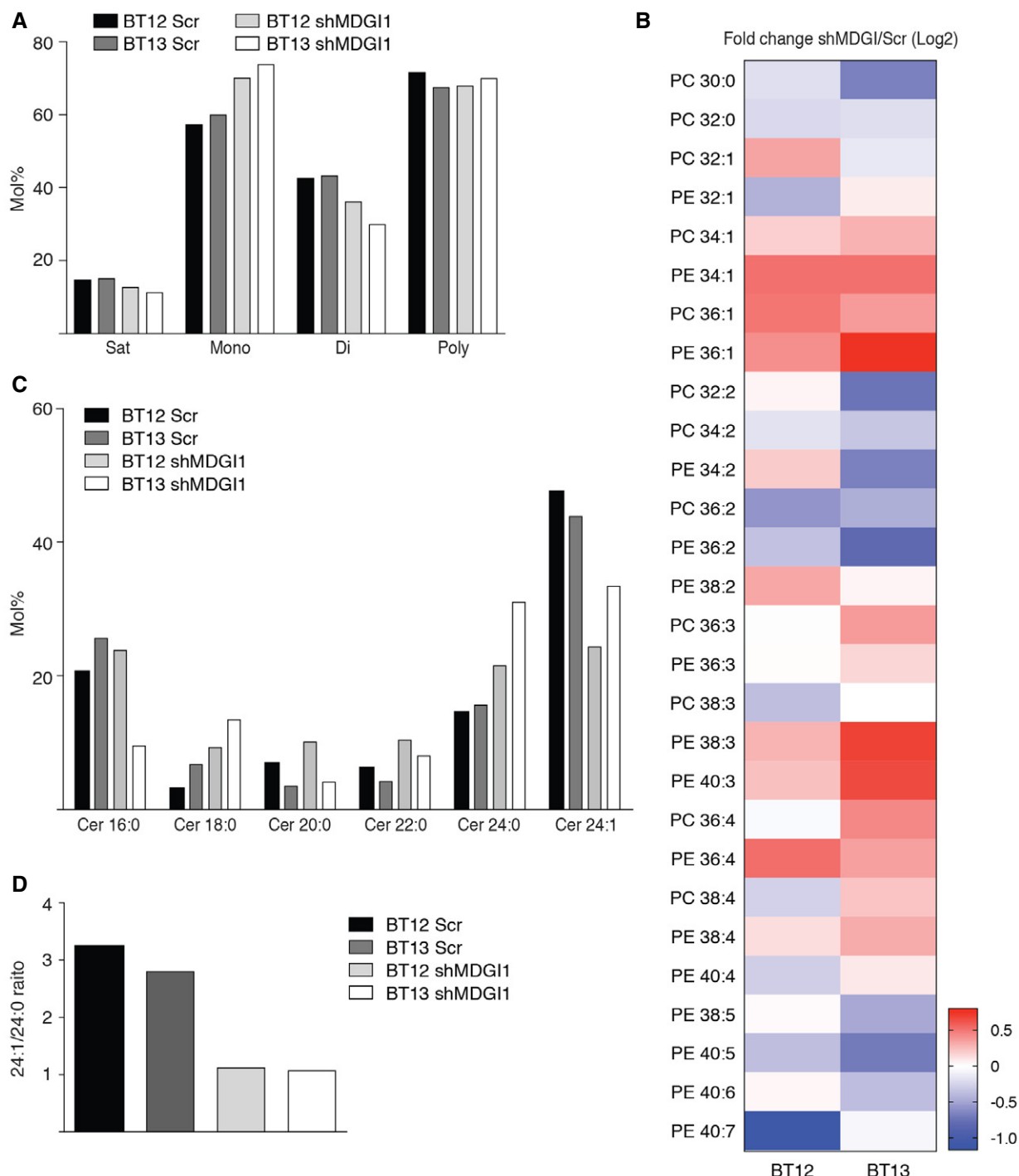

**Figure 5. MDGI silencing induces changes in lysosomal membrane lipid composition.**

A  Proportions (mol %) of saturated (Sat), monounsaturated (Mono), di-unsaturated (Di; total of two double bonds in the acyl chains) and polyunsaturated (Poly; total of 3–7 double bonds in the acyl chains) lipid species from the two main phospholipid classes (phosphatidylcholine PC and phosphatidylethanolamine PE) of the lysosomal fractions of the BT12 and BT13 control (Scr) and MDGI-silenced (shMDGI1) cells. A representative graph of two independent experiments is shown.

B  Fold-change of the main PC and PE species in the lysosomal membranes of MDGI-silenced (shMDGI) and control (Scr) cells (Log2, by mol % data). [Lipid class] [total number of acyl chain carbons]:[total number of double bonds in the chains]. A representative heatmap of two independent experiments is shown.

C  Profiles of ceramide (Cer) species (mol %) in the lysosomal membranes of MDGI-silenced (shMDGI1) and control (Scr) BT12 and BT13 cells. Cer [number of acyl chain carbons]:[number of double bonds in the acyl chain]; all principal species containing a sphingosine chain. A representative graph of two independent experiments is shown.

D  Ratio of the 24:1 to 24:0 (mol %.mol %) Cer species in the lysosomal membranes of MDGI-silenced (shMDGI1) and control (Scr) BT12 and BT13 cells. A representative graph of two independent experiments is shown.

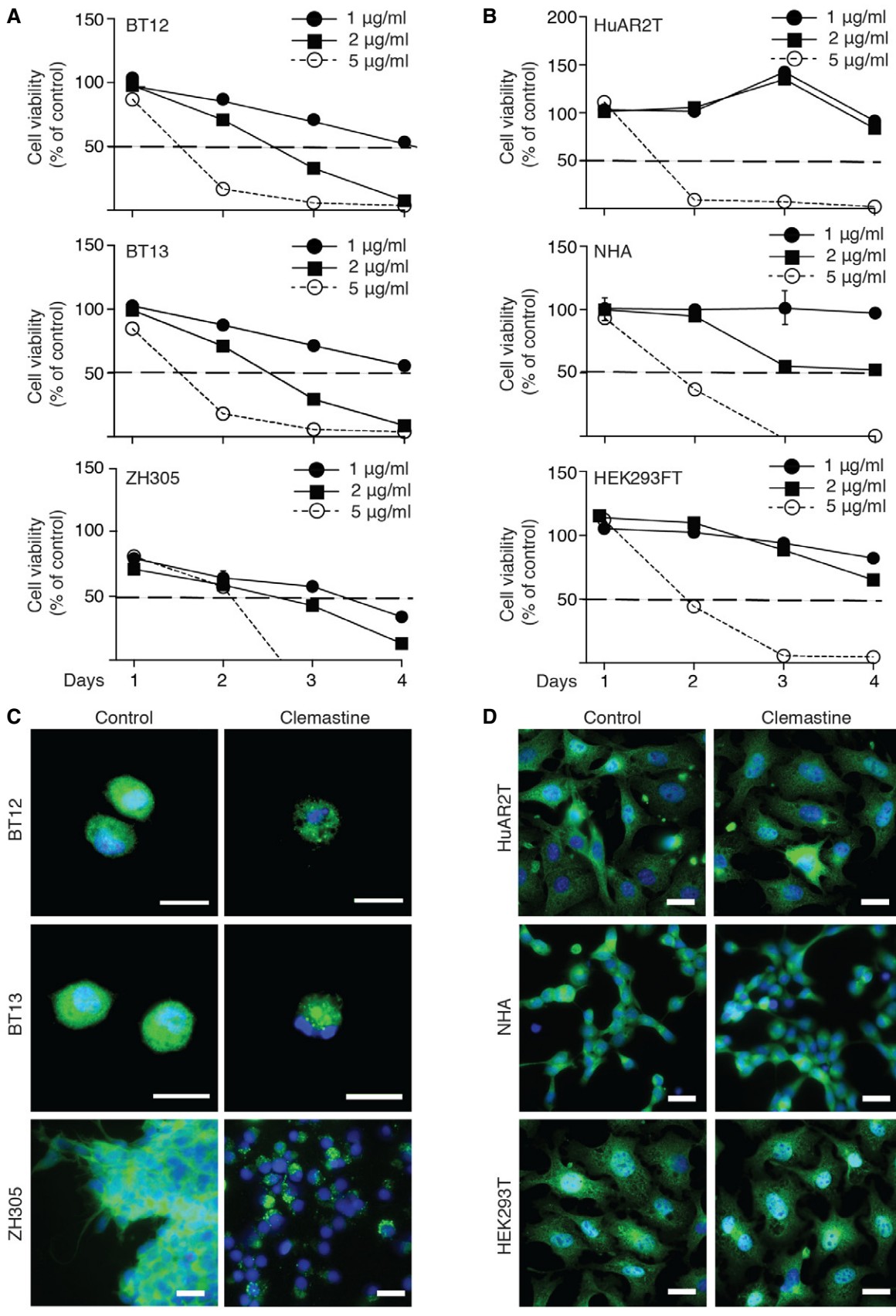

**Figure 6.**

**Figure 6.  Antihistamine treatment induces glioma cell death via lysosomal membrane permeabilization (LMP).**

A  Measurement of the BT12, BT13 and ZH305 glioblastoma cell viability using the MTT assay at the indicated clemastine concentrations and time points ($n = 12$). A dashed line marks the 50% cell viability.

B  Measurement of the viability of normal human endothelial cells (HuAR2T), normal human astrocytes (NHA) and embryonic kidney (HEK293T) cells at the indicated clemastine concentrations and time points ($n = 12$). A dashed line marks the 50% cell viability. Data are represented as mean ± SD.

C  Representative micrographs of BT12, BT13 and ZH305 glioblastoma cells treated with 1 μg/ml of clemastine for 24 h and stained with anti-galectin-1 (LGALS1) antibody (green). Non-treated cells served as control. Nuclei were visualized by using DAPI (blue). Scale bar: 20 μm.

D  Representative micrographs of normal human endothelial cells (HuAR2T), normal human astrocytes (NHA) and kidney cells (HEK293T) treated with 1 μg/ml clemastine for 24 h and stained with anti-LGALS1 antibody (green). Non-treated cells served as control. Nuclei were visualized by using DAPI (blue). Scale bar: 10 μm.

Source data are available online for this figure.

changes in the ceramide FA saturation are plausibly large enough to affect lateral packing and dynamics of the lysosomal membrane. Loss of the unsaturated Cer 24:1 species from the lysosomal membrane leaves behind a membrane enriched with more rigid saturated 24:0, which probably made the membrane less elastic and prone to leakage.

In addition to the impaired membrane integrity and elasticity, activation of Erk in the MDGI-silenced cells could play a role in the process, since phosphorylated Erk [besides being linked to apoptosis induction and cell cycle arrest (Woessmann *et al*, 2002)] has also been shown to enhance cysteine cathepsin expression and activity, sensitizing cells to LMP (Fehrenbacher *et al*, 2008). In addition to the LMP, we also observed decreased activity of the AKT kinase (the master regulator of various cell survival pathways; Chang *et al*, 2003), downregulation of anti-apoptotic proteins XIAP and survivin, which have been linked to caspase-independent apoptosis (Kang *et al*, 2008) as well as an upregulation of the pro-apoptotic serine protease HTRA (Eckelman *et al*, 2006). HTRA contributes to cell death by downregulating XIAP protein levels (Vande Walle *et al*, 2008) and decreases the chemoresistance of colon cancer cells (Xiong *et al*, 2017).

A very recent study by Guishard *et al* showed an evident translational gap in the past clinical trials of glioma: attempts to include drugs that are able to overcome the chemoresistance of glioma stem cells and cross the BBB were not made. However, in the ongoing trials, this has been taken better into account, but there is still room for improvement (Guishard *et al*, 2018). LMP induction has been shown to re-sensitize multidrug-resistant cells to chemotherapy and shown efficacy against apoptosis-resistant cancer cells (Ellegaard *et al*, 2013; Groth-Pedersen & Jaattela, 2013). LMP induction in glioblastoma treatment has not been widely studied, but several studies have recently characterized the mechanism of action of some novel chemotherapeutics and shown that lysosomal dysfunction is a major contributor in drug-mediated cancer cell death (Gonzalez *et al*, 2012; Gutierrez *et al*, 2016; Jiang *et al*, 2016). Among the drugs currently on the market, cationic amphiphilic (CAD) antihistamines induce lysosomal cell death. Moreover, the use of the CAD antihistamines was associated with significantly reduced all-cause mortality among cancer patients when compared with the use of non-CAD antihistamines and adjusted for potential confounders (Ellegaard *et al*, 2016). We chose an older generation CAD antihistamine, clemastine, due to its ability to cross the BBB, to test the sensitivity of our patient-derived glioblastoma cells towards drug-induced LMP. Clemastine is devoid of neurotoxic effects (Liu *et al*, 2016) even though it causes reported fatigue-induced side effects common to many of the antihistaminergic drugs (Verster & Volkerts, 2004). Moreover, clemastine is FDA-approved and currently under clinical evaluation for multiple sclerosis (Green *et al*, 2017). We observed a dramatic loss of the glioblastoma cell viability that was associated with the loss of lysosomal membrane integrity at doses that did not affect the proliferation or viability of several normal cells *in vitro*. When we evaluated the preclinical efficacy of clemastine, the survival of animals bearing intracranial

**Figure 7.  Clemastine treatment eradicates invasive glioma cells.**

A  Representative whole coronal section micrographs of murine brain intracranially implanted with various glioblastoma cells), and daily treated with saline vehicle (Ve) or 50 mg/kg clemastine (Cle, 100 mg/kg single dose administered at day 1) for 12 days starting at day 15 after tumour implantation. BT12 (parental (wt, Ve $n = 9$, Cle $n = 12$) and control shRNA infected (Ve $n = 5$, Cle $n = 5$), ZH305 (Ve $n = 10$, Cle $n = 10$) and BT13 (Ve $n = 10$, Cle $n = 9$). Human glioma cells were visualized with an anti-human vimentin (hVim, red). Nuclei were visualized by using DAPI (white). Scale bar: 1 mm.

B, C  Quantification of the number of the invading single BT12 wt glioblastoma cells (B) and distance of the invaded cells from the primary tumour (C) of vehicle (black bars, Ve) and clemastine-treated (white bars, Cle) animals (Ve $n = 9$, Cle $n = 12$). Data are represented as mean ± SEM. ****$P < 0.0001$, two-tailed, nonparametric Mann–Whitney's *U*-test.

D  Quantification of the number of secondary tumours (diameter > 300 μm) in the murine brain for the indicated treatments and tumour models. BT12 (parental (wt, Ve $n = 9$, Cle $n = 12$) and control shRNA infected (Ve $n = 5$, Cle $n = 5$), and BT13 (Ve $n = 10$, Cle $n = 9$). Data are represented as mean ± SEM. ****$P < 0.0001$, two-tailed, nonparametric Mann–Whitney's *U*-test.

E  Quantification of the number of angiotropic BT12 cells that have co-opted existing blood vessels in the murine brain for the indicated treatments (Ve $n = 9$, Cle $n = 12$). Data are represented as mean ± SEM. ****$P < 0.0001$, two-tailed, nonparametric Mann–Whitney's *U*-test.

F  Quantification of the number of TUNEL-positive cells in the murine brain for the indicated treatments (Ve $n = 9$, Cle $n = 12$). Data are represented as mean ± SEM. ****$P < 0.0001$, two-tailed, nonparametric Mann–Whitney's *U*-test.

G  Representative micrographs of consecutive brain sections (separating distance: 9 μm) showing xenografts treated with vehicle or clemastine and labelled with the anti-human vimentin (red in left panels) and TUNEL (red in right panels). Nuclei were visualized by using DAPI (white). Scale bar: 50 μm.

H  Kaplan–Meier survival curve of the BT12 xenografted animals treated with vehicle (green line, $n = 15$) and clemastine (black line, $n = 12$). Survival of the animals has been measured and pooled from two independent experiments. $P = 0.044$, Mantel-Cox Log-rank test.

Source data are available online for this figure.

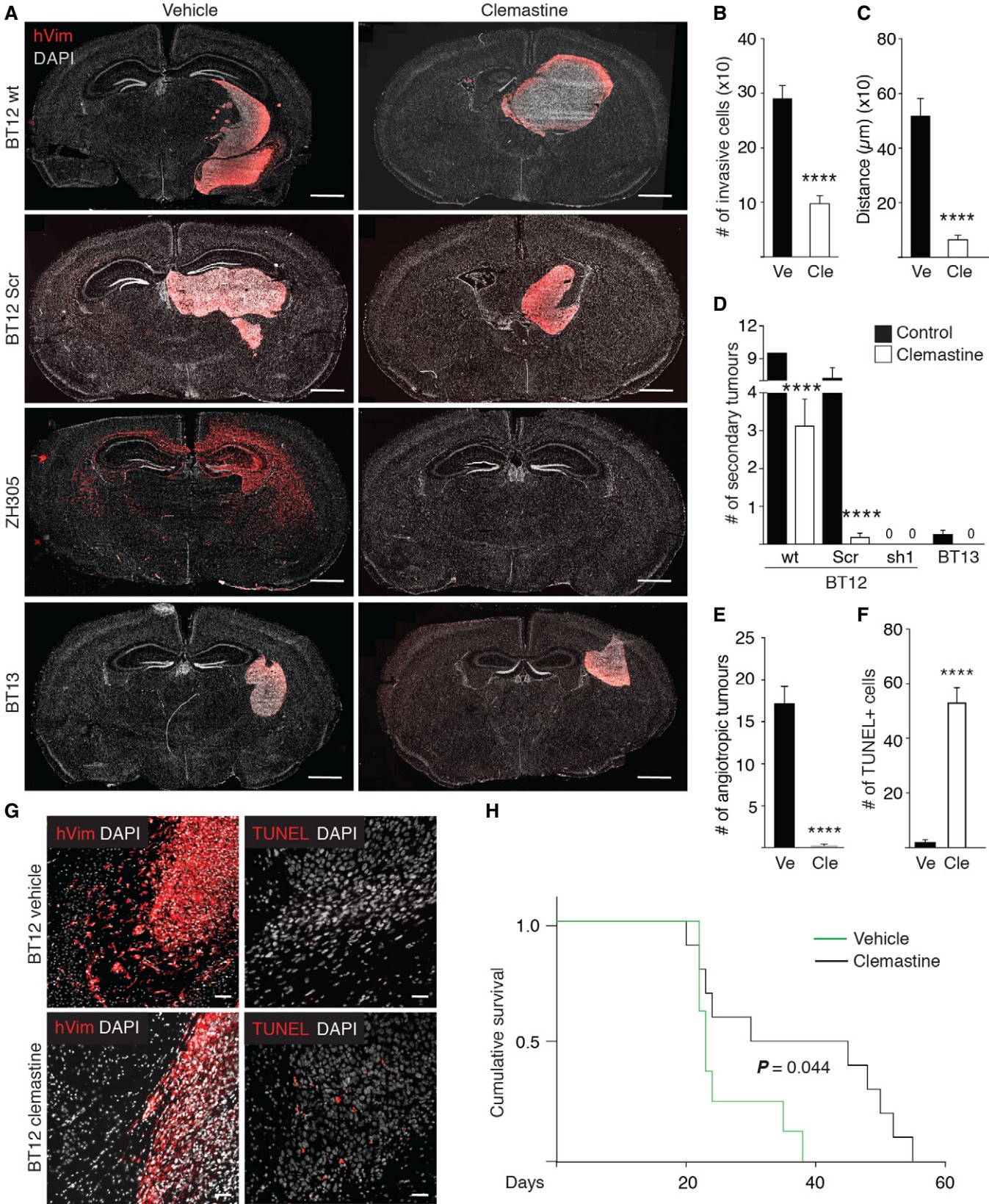

Figure 7.

glioblastoma xenografts was significantly prolonged compared to controls due to the eradication of the invasive glioma cells. Both the co-opted, angiotropic growth and single-cell invasion, as well as collective migration of glioblastoma cells, were inhibited by the systemic delivery of clemastine leading to a nearly complete loss of disseminated cells. The observed cytotoxic effect of clemastine targeted preferentially glioblastoma cells in the vicinity of functional blood vessels and at the leading edge of the primary tumour mass leading to eradication of the secondary tumours. It is known that intratumoural vasculature in glioblastoma is highly aberrant and non-functional (McDonald & Choyke, 2003). Similarly to challenges in delivering systemically administered chemotherapeutics (Bianco *et al*, 2017), clemastine delivery into the primary tumour mass may partially have been compromised due to poorly functional tumour blood vessel network. Lack of clemastine efficacy in reduction of the primary tumour volume may also be due to the dense connective network within the tumour bulk that has been shown to provide exceptional protection of glioblastoma cells to radiotherapy (Osswald *et al*, 2015).

Many triggers of the LMP are known (Cesen *et al*, 2012; Aits & Jaattela, 2013; Appelqvist *et al*, 2013), but genes and cellular pathways that regulate the lysosomal membrane integrity remain poorly understood. Our study demonstrates the crucial role of MDGI in glioma cell survival, linking this fatty acid binding protein to the maintenance of lysosomal membrane integrity. The inoperable, highly invasive and chemo-resistant glioma stem cell-like cells showed an unexpected fragility towards LMP. Thus, our study provides a potential answer to the clinical challenge for the management of the cells that eventually will lead to the recurrence of the disease. These results also suggest re-positioning of an old drug with a new indication. More widely, LMP-inducing agents should be considered as a possible novel treatment option for gliomas.

# Materials and Methods

### Glioma patient cohort

The glioblastoma and grade II–III glioma patient biopsies were obtained from the Department of Pathology, Helsinki University Hospital, Finland. The cohort has been described elsewhere (Joensuu *et al*, 2005; Puputti *et al*, 2006). Briefly, glioblastoma TMA was composed of formalin-fixed paraffin embedded (FFPE) tumours derived from 43 craniotomy patients, subsequently diagnosed with primary glioblastoma between October 2000 and December 2003. The median survival time of the patients was 9 months (range 1–40 months) as calculated from the date of the diagnosis to the date of death. All patients died due to glioblastoma during follow-up. In seven cases, tissue microarray sample was missing or staining was not informative leaving 36 patients in the series.

FFPE grade II–III glioma TMAs were prepared from the tumours of 122 patients who were operated for a primary glioma at the Helsinki University Hospital between 1979 and 2000. In 10 cases, tissue microarray sample was missing or staining was not informative leaving 112 patients for the analysis. Thirty-three of the tumours (30%) were diagnosed as astrocytoma, 16 (14%) as anaplastic astrocytoma, 36 (32%) as glioblastoma, 18 (16%) as oligodendroglioma and 9 (8%) as oligoastrocytoma. The association

of MDGI with clinical and histopathological factors was studied from 76 low-grade samples, excluding glioblastoma patients, and the median follow-up time after surgery was 6.0 years (range 0.6–17.0 years). Ten of the patients lived at the end of the follow-up, while 58 and eight patients died from glioma or other causes, respectively.

### Immunohistochemistry and evaluation of immunostaining

The immunohistochemical stainings using antibodies against SCF/Kit, CD117/KIT-receptor, CD133, EGFR, EGFRvIII, p53 and phosphorylated EGFR (Tyr1173) have been described elsewhere (Joensuu *et al*, 2005; Puputti *et al*, 2006). For immunohistochemical analyses, MDGI expression was detected from the FFPE-samples using TSA kit (Perkin Elmer) according to manufacturer's instructions. The antigen was retrieved with heat treatment in citrate buffer (1.8 mM citric acid, 8.2 mM sodium citrate, pH 6.0). The sections were blocked for 10 min with (10 min with 3% $H_2O_2$ in methanol, 30 min with TNB-buffer (0.1 M Tris–HCl, pH 7.5, 0.15 M NaCl, 0.5% blocking reagent from the TSA kit) and incubated with primary antibody overnight at 4°C and with biotinylated secondary antibody (goat anti-rat immunoglobulin, Dako). The staining was amplified by using SA-HRP and biotinylated tyramide and finally visualized with AEC (0.95 M 3-amino-9-ethylcarbazole, 6% N,N dimethylformamide, 94% sodium acetate, 0.01% $H_2O_2$). The sections were counterstained with Mayer's haematoxylin.

MDGI expression was evaluated blinded to the clinical data and scored as follows: 0, all tumour cells negative; 1, low expression in tumour cells (5–50%); 2, moderate expression in ≥ 50% of tumour cells; and 3, high expression in ≥ 50% of tumour cells. For the statistical analyses, tumours with MDGI expression scored as 0 and 1 were grouped together as well as tumours with MDGI expression scored as 2 and 3. Vascular expression of MDGI was scored as follows: 0, no MDGI-positive vessels; 1, one or more MDGI-positive vessels.

### Patient-derived cells

The glioma patient samples were obtained from surgeries (Kuopio University Hospital, Finland) during years 2010–2011. BT5 cells were obtained from a human gliosarcoma and BT5R cells from its relapse. BT11 cells are from a glioblastoma-containing oligocomponents, and BT12 and BT13 cells are from glioblastoma patients. Informed consent was obtained from all subjects and that the experiments conformed to the principles set out in the WMA Declaration of Helsinki and the Department of Health and Human Services Belmont Report.

Tumours were collected into solution [0.1% BSA (Biowest), 1 M glucose in PBS] centrifuged for 1 min at 250 × *g* and suspended into trypsin-hyaluronidase-kynurenic acid mixture (1.33% trypsin, 0.7% hyaluronidase, 0.2% kynurenic acid, 10,000 U/ml DNase in 1× HBSS solution containing 1 M glucose and 1 M HEPES). Samples were incubated at 37°C for 15–30 min and triturated with a pipette to break down tissues clumps. After filtering through 100-μm nylon mesh filter, the cells were centrifuged for 5 min at 250 × *g*. Cell pellets were suspended into sucrose solution (30.8% glucose in 0.5× HBSS) and centrifuged again for 10 min at 500 × *g*. Cells were suspended in 2 ml of 1× EBSS medium and pipetted on top of

15 ml of BSA-EBSS-HEPES solution (4% BSA, 1 M HEPES in 1× EBSS). After centrifugation for 7 min at 350 × g, the cells were suspended into the complete growth medium. Trypsin, hyaluronidase, kynurenic acid and DNase were obtained from Sigma. HBSS and EBSS were obtained from Gibco and 1 M HEPES-buffer from Lonza.

## Cell culture

The patient-derived brain tumour cells (BT3, BT11, BT12, BT13, BT5, BT5R) were maintained for up to 20 passages in Dulbecco's modified Eagle's medium with Nutrient Mixture F-12 (DMEM/F12, Gibco) supplemented with 2 mM L-glutamine, 2% B27-supplement (Gibco), 100 U/ml penicillin and 100 μg/ml streptomycin, 0.01 μg/ml recombinant human fibroblast growth factor-basic (FGF-b, Peprotech), 0.02 μg/ml recombinant human epidermal growth factor (EGF, Peprotech) and 15 mM HEPES-buffer. The ZH161, ZH305 and S24 were maintained for up to 10 passages in phenol red-free Neurobasal medium (Gibco) supplemented with the same ingredients as described above except 0.02 μg/ml of FGF-b was used. We confirmed by gas chromatography that the medium provided the cells with linoleic acid (18:2n-6) as the main PUFA (50.3 mol % per total FAs). All the cells were maintained and grown at 37°C in a humidified atmosphere containing 5% $CO_2$ unless stated otherwise. Mycoplasma contamination was routinely checked twice a month using the mycoplasma detection kit (11-1050, Minerva Labs).

## Commercial cell lines

The human embryonic kidney cells (293FT, ATCC) were maintained from passages 10–25 in DMEM containing 4.5 g/l glucose, 10% FBS, 1% L-glutamine, 100 U/ml penicillin and 100 μg/ml streptomycin. Human glioma cell line U87MG and murine brain microvessel endothelial cells bEND3 (ATCC) were maintained for not more than 10 and 15 passages, respectively, in DMEM containing 1 g/l glucose, and with the same supplements described above. The generation of stable human U87MG cell lines overexpressing green fluorescent protein (GFP) or MDGI as GFP-fusion (MDGI-GFP) has been previously described (5). The immortalized human umbilical vein endothelial cells (HuAR2T) were obtained from the Ojala Lab (University of Helsinki, Finland) and maintained from passages 15–25 in Endothelial Basal Medium 2 (C-22211) added with Supplements Pack Endothelial Cell GM2 (C-39211; Promocell) and 2 μg/ml doxycycline.

## Lentivirus production and generation of MDGI-silenced cells

The MDGI/FABP3 shRNA-constructs in the pLKO.1 vector were obtained from the RNAi Consortium shRNA library (Broad Institute of MIT and Harvard). The sequences were as follows: shMDGI1: 5′GACCAAGCCTACCACAATCAT3′, shMDGI2: 5′GACAGGAAGGTCAAGTCCATT3′.

Second-generation lentiviruses were produced by co-transfecting the MDGI shRNA or Scr-containing plasmids and lentiviral packaging plasmids CMVg and CMVΔ8.9 (Addgene) together with Fugene6-transfection reagent (Promega) into 293FT cells. After 72 h, the virus-containing supernatants were collected and filtrated. On the day of transduction, human glioma cell spheres were dissociated with Accutase (Gibco) and transduced with viral supernatants together with 8 μg/ml Hexadimethrine bromide (Sigma). On the next day, the cells were supplied with the fresh complete medium. MDGI silencing was verified using Western blot and real-time qRT–PCR.

## Immunofluorescence

In order to attach the suspension cells, glass coverslips were first coated with Poly-D-Lysin (Sigma) according to manufacturer's instructions. Cells were cultured for 24 h after which they were fixed with 4% PFA and permeabilized with 0.5% NP-40 in PBS. The cells were blocked with 3% BSA-PBS and incubated with the primary and fluorescently labelled secondary antibodies (Alexa Fluor, Life technologies). Nuclei were visualized with DAPI (Vectashield, Vector Laboratories).

## Western blot

The cells were lysed either in RIPA-buffer [150 mM sodium chloride, 50 mM Tris, 2 mM EDTA, 0.5% sodium deoxycholate, 0.1% sodium dodecyl sulphate SDS, 0.5 mM DTT, Complete ultratablet protease inhibitor, (Roche)] or in SDS-buffer (150 mM Tris, pH 6.8, 1.2% SDS, 30% glycerol, 15% β-mercaptoethanol). Protein concentrations of the extracts were determined spectrophotometrically (Pierce BCA Protein Assay Kit, Thermo Scientific). 10–20 μg of protein was loaded to a 12% SDS–PAGE gel followed by a transfer to PVDF membrane (Immobilon, Millipore) either by electroblotting in transfer buffer (20% MetOH, 250 mM Tris, 1.9 M glycine) 100 V for 1 h at 4°C or by Transblot Turbo device (Bio-Rad) using manufacturer's instructions. After blocking (5% BSA in 0.1% TBS-Tween), the membrane was probed with the primary antibodies o/n at 4°C. After several washes, the membrane was probed with horseradish peroxidase-conjugated secondary antibodies, washed again and finally visualized using the SuperSignal West Pico kit (Thermo Scientific) or with Clarity Western ECL (Bio-Rad).

## Quantitative real-time PCR

RNA extraction was performed using the Nucleospin RNA II kit (Macherey-Nagel) and reverse-transcribed into complementary DNA (cDNA) with High-capacity cDNA Reverse Transcription Kit and MultiScribe Reverse Transcriptase (Applied Biosystems). Quantitative real-time PCR (qRT–PCR) was performed using SYBR Green PCR Mix (Kapa Biosystems). Relative gene expression was normalized to ADP-ribosylation factor 1 (ARF1) housekeeping gene. Relative quantification of the expression levels between the samples was performed according to the Delta Ct method ($\Delta C_t$). Primer sequences in 5′-3′-direction were as follows: MDGI (fwd): CCTGGAAGCTAGTGGACAGC; MDGI (rev): TAGCAAAACCCACACCGAGT; EGFR (fwd): TCCAGTGGCGGGACATAGT; EGFR (rev): TGGATCACACTTTTGGCAGC; ARF1 (fwd): TCCCACACAGTGAAGCTGATG; ARF1 (rev): GACCACGATCCTCTACAAGC.

## MTT-proliferation assay

5,000 cells/well were plated on 96-well plates in 3–10 replicates (100 μl volume). At the indicated time points, 10 μl of 3-(4,5-

Dimethylthiazol-2-yl)-2,5-Diphenyltetrazolium Bromide (MTT; 5 mg/ml in PBS) was added and the cells were incubated for 2 h. Finally, the cells were lysed (10% SDS, 10 mM HCl) o/n and the absorbance was measured at 540 nm using Multiskan Ascent software version 2.6 (Thermo Labsystems).

### Cell viability and apoptosis

Trypan blue exclusion method was used to estimate the cytotoxic effects of MDGI silencing. Cells were diluted 1:2 in 0.4% Trypan blue solution (Sigma), after which viable and dead cells were counted. The percentage of viable cells was determined as follows: (number of viable cells/total cell number) × 100. The proportions of apoptotic cells were determined with Annexin V staining which binds to phosphatidylserine residues translocated to the outer cell membrane during early stages of apoptosis. Cells were dissociated with Accutase and stained with Annexin V conjugate (Annexin V Alexa Fluor 488, Molecular Probes), for 15 min in RT. After washes with binding buffer (10 mM HEPES—140 mM sodium chloride—2.5 mM $CaCl_2$), the lentiviral transduced cells were fixed with 4% PFA (containing 2.5 mM $CaCl_2$) for 10 min at 4°C, suspended into binding buffer and analysed with flow cytometer (BD Accuri C6) equipped with appropriate lasers. Cell gating and data analysis was performed using FlowJo v10.1 software (Tree Star Inc.).

### Apoptosis arrays

Expression of apoptosis-associated proteins was determined using the Human apoptosis antibody array (ab134001, Abcam) containing antibodies against 43 proteins in duplicates and additional reference spots. Membranes were incubated with 335 µg of cell extracts, immunoblotted and exposed to X-ray films (Super RX, Fuji Medical) according to manufacturers' instructions. Pixel densities of each protein spot were determined by using the ImageJ software.

### EGFR inhibition

Glioma cells ($5 × 10^3$) were plated on 96-well plates in triplicates and treated with various concentrations (0.1–10 µM) of gefitinib (InvivoGen, USA). After 48 h MTT was added, cells were lysed and the absorbance was measured as described above. Half-maximal inhibitory concentration ($IC_{50}$) was determined for each cell line using the following formula: $IC_{50}$ = (50%–Low Inh%)/(High Inh%–Low Inh%) × (High Conc–Low Conc) + Low Conc, where: Low Inh%/High Inh% = % inhibition directly below/above 50% inhibition; Low Conc/High Conc = Corresponding concentrations of the test compound.

### Colony forming cell assay in methylcellulose

Methylcellulose stock solution (3%) and protocol were obtained from the R&D Systems. Briefly, $1 × 10^4$ cells were suspended as single cells in complete medium containing the growth factors EGF and FGF-b and semi-solid matrix Methylcellulose stock solution (final concentration 1.3%). One millilitre of the final cell mixture was added into 35-mm plate in triplicates and incubated for 3 weeks at 37°C containing 5% $CO_2$. The colonies were imaged using Nikon Eclipse Ti-E inverted microscope and quantified using the

CellProfiler cell image analysis software (http://cellprofiler.org/citations/). The experiment was repeated three times.

### Anchorage-independent growth assay

Cells (3,000/35-mm well, triplicates) were suspended in medium containing 0.35% agarose and pipetted into 6-well plates containing a 2-ml layer of solidified 0.7% agar in the medium. Complete medium was added twice a week. After 14 (U87MG) or 27 (BT12 and BT13) days, eight images per well were taken (Zeiss Axiovert) and the number of colonies was quantified using the ImageJ software (National Institute of Health, USA). The experiment was repeated twice.

### Hypoxia and serum starvation

For the hypoxia treatment, $3 × 10^5$ cells were plated on 35-mm dishes 1 day before the experiments. The cells were incubated at 37°C in a hypoxia chamber (Ruskinn Technology Limited, UK) in 1% of $O_2$ and 5% of $CO_2$ for 24 h. For the FBS treatments, $3 × 10^5$ cells were plated on 35-mm dishes and grown for 7 days in DMEM containing 10% FBS or in serum-free DMEM/F12. Afterwards, the cells were washed with ice-cold PBS, lysed and analysed with Western blot.

### Ex vivo brain slice cultures

Female FVB mice (5–8 weeks of age) were anesthetized, and their brains were removed. Cerebellum was excised, and the two hemispheres were embedded into 4% Low Melting Point Agarose (Thermo Scientific) in PBS. Slices of 500 µm were cut with a vibratome (HistoLab) and placed on top of 0.4-µm filter membranes (Millipore) in 6-well plates. Neurobasal A (1 ml, Gibco) supplemented with 1 mM glutamine, 2% B-27 (Gibco) and 1% penicillin/streptomycin was added to the bottom of each well. The medium was changed every other day. The slices were incubated for 1 week (37°C, 5% $CO_2$) before starting the experiments (Ohnishi et al, 1998). The spheres were formed o/n in U-shaped 96-well plates coated with 0.6% agarose (4,000 cells/well), after which they were placed to the brain slices. The GFP-fluorescent spheres were imaged at indicated time points by taking 10-µm optical slices with Leica TCS SP2 confocal microscope. Image stacks were generated using the ImageJ software, and the number of invaded and migrated cells/cell groups was quantified manually. The experiment was repeated twice with 3–5 individual spheres/cell line/slice in each experiment.

### Analysis of lysosomal membrane permeabilization

LMP was determined using the LGALS1 puncta-staining assay (Aits et al, 2015). BT12 cells treated with 2 mM L-leucine O-methyl ester (LLOMe, Santa Cruz Biotechnology) were used as positive controls for LMP. For the quantification of LGALS1 staining (cytoplasmic/puncta), DAPI- and LGALS1-stained cells grown on PDL-coated coverslips were scanned with 3DHISTECH Pannoramic P250 FLASH II using Zeiss Plan-Apochromat objective (20×/NA 0.8) and pico.Edge 4.2 CMOS camera. The number of LGALS1-stained cells (cytoplasmic/puncta) and unstained dead cells was manually calculated from each coverslip ($n$ = 6, 50 $mm^2$ area) using Panoramic viewer software (3DHISTECH).

## Cytoplasmic Cathepsin B activity measurement

Protocol was modified from Jaattela and Nylandsted (2015). Cells ($2 \times 10^4$) in 100 μl of complete medium and growth factors were plated in poly-D-lysine-coated 96-well plates in quadruplicates and incubated o/n. Cytosolic and total cell extracts were prepared by incubating the cells in digitonin extraction buffer on ice for 15 min on a rocking table. Used digitonin concentrations for cytosolic and total cell extractions were 20 μg/ml and 200 μg/ml, respectively. After transferring the extracts into a new 96-well plate, the cells were lysed in SDS-lysis buffer and protein concentrations were determined using the Pierce BCA Protein Assay Kit (Thermo Scientific) according to the manufacturer's instructions. To measure the cathepsin B activity, 50 μl of cytosolic and total extracts was mixed with 50 μl of reaction buffer containing cathepsin B substrate (Calbiochem) in a black 96-well plate. The plate was preincubated at 30°C for 5 min, and then, the kinetics of cathepsin B activity was measured using FLUOstar Omega Microplate Reader (BMG Labtech) at 30°C for 20 min (Ex 380 nm; Em 460 nm; 45-s interval). To calculate the cytosolic cathepsin B release, measured enzyme activities were first normalized to the corresponding protein concentration of the same well, and then, cytoplasmic activity was compared to the total cellular activity. The experiments were started at day 3 post-lentiviral transduction and repeated three times.

## Pan-cathepsin inhibition assay

For cathepsin inhibition assays, $5 \times 10^3$ cells in 100 μl of complete medium and growth factors were plated in 96-well plates in triplicates. Cells were treated with 0, 1 or 2 μM of pan-cathepsin inhibitor K777 (AdipoGen Life sciences). At indicated time points, MTT was added, cells were lysed, and the absorbance was measured as described above. Fresh inhibitor was added to the cells every other day during the experiments. The experiment was started at day 4 post-lentiviral transduction and repeated three times.

## Lysosomal extracts

Lysosomal extracts were prepared at day 6 post-lentiviral transduction by density gradient ultracentrifugation using the Lysosome Enrichment Kit for Tissue and Cultured Cells (Thermo Scientific) according to the manufacturer's protocol. One readout constituted of six independent spheroid culture samples pooled together.

Presence of the lysosomes and purity of the collected fractions were verified using Western blot analysis with specific antibodies against lysosomes (LAMP2) and endoplasmic reticulum (Calnexin 1). Extracted lysosome pellets were stored at −80°C and further analysed by mass spectrometry. The experiment was performed two times.

## Mass spectrometric analysis of lysosomal lipids

Total lipids in the lysosomal fractions were extracted using the Folch method (Folch et al, 1957). The lipids were identified and quantified by direct infusion electrospray ionization-tandem mass spectrometry as previously described (Tigistu-Sahle et al, 2017) with Agilent 6490 Triple Quad LC/MS with iFunnel technology (Agilent Technologies, Inc.). In brief, the lipid extracts in chloroform: methanol (1:2, v:v) were spiked with 10 internal standards and

supplemented with 1% $NH_4OH$ just prior infusion into the MS at a flow rate of 10 μl/min. The lipids were detected by using lipid-class specific detection modes (e.g. precursor ion m/z 264 for ceramide, m/z 184 for PC and neutral loss scan of 141 amu for PE; Brugger et al, 1997). The spectra were processed by MassHunter Workstation qualitative analysis software (Agilent Technologies, Inc.), and the individual lipid species were quantified using the internal standards and free software called Lipid Mass Spectrum Analysis (Haimi et al, 2006). The results are described as mol % of each molecular species in its lipid class and marked as the number of acyl carbons: number of double bonds (e.g. 36:2).

## Clemastine treatment

Clemastine fumarate salt (Sigma) cytotoxicity was tested at the indicated concentrations and time points using the MTT protocol as described above. LMP induction by clemastine was verified using the LGALS1 puncta-staining assay described in the previous section (Aits et al, 2015). Cells were treated with DMSO or clemastine (1 μg/ml) at 37°C o/n and then lysed in RIPA-buffer for Western blot analyses.

## Animal experiments

Animal experiments were approved by the Committee for Animal Experiments of the District of Southern Finland (ESAVI/6285/04.10.07/2014). Six-week-old immunocompromised NMRI-nu (Rj: NMRI-Foxn1[nu]/Foxn1[nu], Janvier Labs) mice were intracranially engrafted using a stereotaxic injector (World Precision Instruments) in the corpus callosum with $10^5$ U87MG-GFP ($n = 5$) or U87MG-GFP-MDGI ($n = 9$) or LN229-GFP ($n = 10$) or LN229-GFP-MDGI ($n = 10$) cells in 5 μl of PBS under ketamine and xylazine anaesthesia. Post-operative painkiller (temgesic) was locally administered for 2 days. Following the same methodology, the clemastine preclinical study included the implantation of $10^5$ cells dissociated from spheroids in 5 μl of PBS. The cohorts were distributed as BT12 parental (wt, Ve $n = 9$, Cle $n = 12$), shMDGI infected (Ve $n = 5$, Cle $n = 5$), control shRNA infected (Ve $n = 5$, Cle $n = 5$), ZH305 (Ve $n = 10$, Cle $n = 10$) and BT13 (Ve $n = 10$, Cle $n = 9$). Two weeks after tumour implantation, animals were randomized and clemastine (200 μl in PBS, intraperitoneally) was injected daily at the indicated doses for a duration of 12 days in one cohort or until the physical manifestation of tumour burden for the survival studies. Control animals were treated under the same modalities with the vehicle (200 μl). At the endpoint of the experiment, animals were euthanized and brains were snap-frozen in −50°C isopentane (Honeywell) until tissue processing.

## Animal tissue processing and analysis

Snap-frozen xenografted brains were cut using a cryotome (Cryostar NX70, Thermo Scientific) into series of 9-μm-thick coronal sections collected from the frontal to the anteroposterior part of the tumour. Tissue sections were fixed in 4% PFA, blocked with 5% FBS and 0.03% Triton X-100 (Sigma) and stained for the human vimentin or CD31 and MDGI by immunofluorescence. Whole-brain sections were then scanned using a slide scanner (3DHistec). When required, verification of co-located staining was performed on 9-μm distant

**The paper explained**

**Problem**

Glia-derived brain tumours are the most frequent primary central nervous system cancers. In particular, its deadliest and the most frequent form, glioblastoma, is highly invasive cancer. Glioblastoma cells cannot be completely removed by surgery, and they exhibit high resistance to therapies. Finding a potential weakness of these invasive glioblastomas is required to improve the otherwise very poor patient prognosis. Current drugs suffer from poor specificity, which increases the off-target and side effects, and even poorer brain delivery due to the presence of a highly selective blood–brain-barrier (BBB). This study investigated a novel modality of the glioblastoma cell death, the lysosomal membrane permeabilization (LMP). It identifies the heart-type fatty acid binding protein (MDGI/FABP3) as a central regulator of the lysosomal membrane integrity in glioblastoma cells. The LMP can also be induced *in vivo* by administering drugs that can pass through the BBB, such as the cationic amphiphilic (CAD) antihistamine clemastine.

**Results**

Using patient databases, tissue samples and live glioblastoma cells isolated from several patients, we show that MDGI is a biomarker of tumour invasiveness. Moreover, silencing of MDGI provoked death of glioblastoma cells through activation of the LMP. Using high-throughput lipidomics analyses, we were able to pinpoint the accumulation or depletion of very specific lysosomal phospholipids in the absence of MDGI. We linked the lysosomal membrane destabilization induced by the silencing of MDGI to interrupted transportation of linoleic acid from the extracellular space to the intracellular compartment. Among the family of lysosomotropic LMP-inducing agents, the antihistamine clemastine was able to kill patient-derived glioblastoma cells without compromising the viability of non-cancer cells. Ultimately, the administration of clemastine to mice bearing patient-derived tumours showed a very significant reduction of the invasive cancer cell density. Depending on the patient-isolated cells, clemastine treatment left only a very circumscribed tumour or completely eradicated highly diffusive glioblastoma.

**Impact**

Bearing in mind, the chemo-resistant nature of infiltrating gliomas and the poor delivery of conventional drugs due to the BBB, the repurposing of antihistamines and other CADs to trigger the LMP in invasive neoplasms should be considered as an adjuvant therapy to the standard of care, i.e. the surgical resection of the primary tumour and conventional radio- and chemotherapies.

consecutive sections of each brain. Primary tumour volume determination was obtained by measuring the tumour area of 100-µm distant consecutive sections covering the whole tumour (up to 30 histologic sections). Quantification of the different tumour parameters (tumour cell invasion, number of angiotropic satellites) was performed on 10 sections equally distributed along the entire tumour.

**Microscope imaging**

The samples were imaged with a fluorescent upright microscope (Zeiss Axioplan and Axioimager) using AxioVision software. For the light microscopy, the cells were visualized and imaged with an inverted epifluorescence microscope (Zeiss Axiovert 200) equipped with AxioVision software. Confocal images were taken with Zeiss LSM 780 and 880 equipped with appropriate lasers.

**Patient study approval**

The ethics committee of the Hospital District of Helsinki and Uusimaa approved this study. The Ministry of Social Affairs and Health, Finland, permitted the use of tumour tissue. The ethics committee of the Pohjois-Savo Health Care District municipalities permitted the use of human glioma tissue specimens (53/2009, 1.8.2009-31.7.2019).

**Statistics**

Association of MDGI expression with other factors was tested using a Chi-squared test or Fisher's exact test and Mann–Whitney's *U*-test was used for the MDGI vs. age analysis. All tests were 2-tailed. The cumulative survival rates of patients were estimated by using the Kaplan–Meier method, and survival between groups, hazard ratio (HR) and their 95% confidence interval (95% CI) were computed by using the Cox proportional hazards model. Overall survival was calculated from the date of the surgery of primary tumour to death, censoring patients still alive on the last date of follow-up. Glioma-specific survival was calculated from the date of the surgery of primary tumour to glioma caused death, censoring patients still alive or lost from follow-up for other cause on the last date of follow-up. Analyses were conducted with the IBM SPSS 22 software. Results from all other experiments were analysed using the GraphPad Prism 7 software (La Jolla, USA) using two-tailed, nonparametric Mann–Whitney's *U*-test. All the experiments were repeated at least three times with triplicates unless stated otherwise. For the lipidomics, two cell lines were used and the experiment was repeated two times. The exact *P*-values are listed in the Appendix Table S3.

**Expanded View** for this article is available online.

## Acknowledgements

This work was supported by grants from the Ida Montin, Oskar Öfflund and K. Albin Johansson foundations as well as from the Finnish Cancer Organizations, Jane & Aatos Erkko Foundation and Sigrid Juselius Foundation. P Filppu and M. Hyvönen have been supported by the Doctoral Programme in Biomedicine. The Biomedicum Imaging Unit is acknowledged for their assistance in microscopy imaging, Genome Biology Unit (Faculty of Medicine, University of Helsinki, Biocenter Finland) for the 3DHISTECH Pannoramic P250 FLASH II digital slide scanner service. The Biomedicum Functional Genomics Unit (FUGU; Faculty of Medicine, University of Helsinki, HiLIFE) is acknowledged for providing the glycerol stocks corresponding shRNAs. Drs. Outi Rautsi and Kirsi Vuorinen are acknowledged for isolating the patient-derived glioma cells.

## Author contributions

Conception and design: VLJ, PF, MaH, KL, PL; development of methodology: VLJ, MiH, PF, SPT, IB; acquisition of data (provided animals, acquired and managed patients, provided facilities, etc.): VLJ, MaH, PF, SPT, OT, JJ, MiH, RK; analysis and interpretation of data (e.g. statistical analysis, biostatistics, computational analysis): HS, HJ, VLJ, PF, MaH, SPT, MiH, RK; writing, review and/or revision of the manuscript: VLJ, PF, MaH, SPT, HS, HJ, OT, MW, KL, MiH, RK, PL; administrative, technical or material support (i.e. reporting or organizing data, constructing databases): HS, HJ, OT, JJ; study supervision: VLJ, IB, MiW, KL, RK, PL.

## Conflict of interest

The authors declare that they have no conflict of interest.

## For more information

(i)  http://gliovis.bioinfo.cnio.es

(ii)  The Ivy_GAP dataportal: http://glioblastoma.alleninstitute.org

(iii)  https://cellprofiler.org/

(iv)  Laakkonen lab web-site: https://www.helsinki.fi/en/beta/tumor-progression-and-metastasis

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
