## [Review Process File · EMBO Molecular Medicine]

Vulnerability of invasive gliomablastoma cells to lysosomal membrane destabilisation

Vadim Le Joncour, Pauliina Filppu, Maija Hyvönen, Minna Holopainen, S. Pauliina Turunen, Harri Sihto, Isabel Burghardt, Heikki Joensuu, Olli Tynnen, Juha Jääskeläinen, Michael Weller, Kaisa Lehti, Reijo Käkälä and Pirjo Laakkonen

Review timeline:

Submission date:	23 February 2018
Editorial Decision:	5 April 2018
Authors' response:	14 November 2018
Editorial Decision:	13 Decemeber 2018
Revision received:	9 March 2019
Editorial Decision:	21 March 2019
Revision received:	3 April 2019
Accepted:	5 April 2019

Editor: Lise Roth

Transaction Report:

1st Editorial Decision

5 April 2018

Thank you for the submission of your manuscript to our editorial office. We have now heard back from the three referees whom we asked to evaluate your manuscript.

As you will see, the referees all acknowledge the potential interest and clinical relevance of the study, however they also have serious and overlapping concerns that preclude further consideration of the article at this time. They realize that addressing these comments would require a lot of additional work, time and effort.

As clear and conclusive insight into a novel clinically relevant observation is key for publication in EMBO Molecular Medicine, and together with the fact that we only accept papers that receive enthusiastic support upon initial review, I am afraid that we cannot offer to consider the manuscript further.

Given the potential interest and novelty of the findings, we would, however, be willing to consider a new manuscript on the same topic if at some time in the near future you obtained data that would considerably strengthen the message of the study and address the referees concerns in full. To be completely clear, however, I would like to stress that if you were to send a new manuscript this would be treated as a new submission rather than a revision and would be reviewed afresh, in particular with respect to the literature and the novelty of your findings at the time of resubmission. If you decide to follow this route, please make sure you nevertheless upload a letter of response to the referees' comments.

I am sorry that I could not bring better news this time and hope that the referee comments are helpful in your continued work in this area.

***** Reviewer's comments *****

Referee #1 (Remarks for Author):

The manuscript is mainly focused on the role of MDGI in glioblastoma. The authors show a novel functional role of MDGI in glioma cell invasion, survival and maintenance of the lysosomal membrane integrity (LMP). They also demonstrate an unexpected sensitivity of glioma cells to an LMP-inducing drug, anti-histamine clemastine in vitro and in patient-derived xenografts orthotopically implanted.

The results are very interesting, the authors propose LMP-inducing agents as possible novel options for GBM treatment, but the data are still very preliminary.

The manuscript is mainly focused on the role of MDGI in glioblastoma. Although, it is difficult to follow the flow of information specially in the second part of the manuscript since the authors move on discussing the treatment of GBM PDX mice with LMP-inducing drug not addressing experimentally the MDGI regulation of the lysosomal membrane integrity. More additional experimental work is then necessary.

The manuscript is very descriptive and the main findings are not explored in depth and not discussed.

Another general comment: the authors use a terminology a little bit unusual and confusing for the GBM field (patient-derived BT12...tumoroids instead of CSCs or TICs) and the in vitro model system is not characterized in terms of tumorigenicity.

They should better focus the manuscript on MDGI and maintenance of the lysosomal membrane integrity. The title doesn't reflect the main results.

- 1) All western blots should be repeated in order to load an equal amount of proteins. They should add a relative quantification of the protein of interest.
- 2) In Fig2 the authors should explain the meaning of the relative number of satellite tumors and how they measured it.
- 3) To measure self-renewal ability of CSCs they should use methylcellulose assay and/or in vitro limiting dilution assay.
- 4) For in vivo studies, they have used only one patient-derived cell line. Considering the high heterogeneity of GBM among patients, other tumor-derived cell lines should be used.
- 5) They do not investigate in vivo for possible side effects of clemastine treatment.
- 6) The authors should repeat the evaluation of clemastine treatment in PDXs derived from the injection of MDGI silenced cells.

Referee #2 (Comments on Novelty/Model System for Author):

Authors have utilized very relevant patient-derived models.

Referee #2 (Remarks for Author):

In this manuscript Joncour et. al. have identified FABP3 as novel biomarker of glioma cell invasion. In addition they also demonstrate that FABP3 silencing leads to glioma cell death via induction of LMP. Finally, using LMP inducing small molecule they demonstrated that LMP inducing drug significantly prolong survival of intracranial glioma bearing animals. This is extremely novel study with high translational impact, given that LMP inducing drug used in the study is already approved for use in humans. While authors have presented comprehensive data in support of their findings some additional data will strengthen their findings and will improve the experience for avid scientific readers.

Suggestions:

1. Authors are requested to provide example IHC images representative of each staining score.
2. Authors are requested to look at the public databases (such as TCGA) and compare FABP3 expression level and survival of glioma patients.
3. Authors are requested to look at the pharmacodynamics effects of Clemastine in intracranial xenografts, aka, do clemastine treated tumors in vivo demonstrate increase in LGALS1-puncta?

4. An experiment with cathepsin inhibitor to rescue LMP induced cell death downstream of FABP3 knockdown and clemastine treatment will comprehensively validate authors claim of MOA.

Referee #3 (Remarks for Author):

The study by Le Joncour et al. investigated the roles of mammary-derived growth inhibitor (MDGI) and lysosomal membrane destabilization in gliomas. The study shows that MDGI expression correlates with survival in grades II-III gliomas but not in glioblastoma (GBM). MDGI expression did not correlate with glioma grade. Overexpression of MDGI in the GBM cell line U87 led to increased invasion in vitro and in vivo. Conversely, silencing of MDGI in two patient-derived GBM cell lines led to inhibition of cell viability in a caspase independent manner but via lysosomal membrane permeabilization (LMP). Use of the LMP-triggering (anti-histaminic) drug clemastine inhibited cell viability and xenograft growth and prolonged animal survival.

The data presented in this manuscript are novel and potentially significant as very little is known about MDGI in gliomas and the use of LMP-triggering clemastine has not been tested as experimental GBM therapy before. The manuscript is clearly written. However, presented data are somewhat limited and superficial and the expression, functional and therapeutic data are disconnected. Substantial additional work is required to confirm and focus the overall message of the study.

Major critique:

- The shown survival data are in lower grade gliomas, but all functional and therapeutic data are in GBM (there was no association with survival of MDGI in GBM). TCGA data should be analyzed for potential association with GBM survival or with glioma grade of MDGI. TCGA or other public data should also be analyzed for a potential differential expression of MDGI in GBM relative to either lower grade gliomas or normal brain .
- The overexpression of MDGI is done in one cell line only. The endpoint is invasion in vitro and in vivo. In vitro data in a second cell line should be generated. Also, why was tumor growth or animal survival not assessed in vivo? If MDGI is an important regulator of malignancy, its overexpression should affect in vivo xenograft growth and/or animal survival.
- How does MDGI regulate LMP? This issue is mentioned in the discussion but not experimentally addressed. Mechanistic experiments that uncover the mode of action of MDGI would strengthen the findings.
- The clemastine data are not directly related to MDGI. Is the clemastine effect mediated by MDGI inhibition? That would be worthy of investigating. If not, the effect of MDGI silencing on in vivo tumor growth should be tested.

Minor critique:

- In which cell line(s) were the data shown in figure 4 performed. This is not mentioned in the text or in the legend.
- "Instabilization" is not an English word. "Destabilization" or "instability" should be used instead.

Authors' response

14 November 2018

Referee #1 (Remarks for Author):

The manuscript is mainly focused on the role of MDGI in glioblastoma. The authors show a novel functional role of MDGI in glioma cell invasion, survival and maintenance of the lysosomal membrane integrity (LMP). They also demonstrate an unexpected sensitivity of glioma cells to an LMP-inducing drug, anti-histamine clemastine in vitro and in patient-derived xenografts orthotopically implanted. The results are very interesting, the authors propose LMP-inducing agents as possible novel options for GBM treatment, but the data are still very preliminary. Another general

comment: the authors use a terminology a little bit unusual and confusing for the GBM field (patient-derived BT12...tumoroids instead of CSCs or TICs) and the in vitro model system is not characterized in terms of tumorigenicity.

As suggested by the Referee and to comply with the terminology of the field, we have changed the terminology in the resubmitted manuscript to patient-derived spheroids instead of tumoroids. The tumorigenicity characterisation of all our cell lines was performed prior to this study. Each patient-derived cell line was implanted in animals, tumour growth was evaluated, and carefully dissected for more specific features such as the invasive potential, type of angiogenesis, recruitment of pericytes, etc. However, only the relevant tumorigenicity characteristics for the study of MDGI are presented in the manuscript.

They should better focus the manuscript on MDGI and maintenance of the lysosomal membrane integrity. The title doesn't reflect the main results.

*This comment was also presented by Referee #3. To understand how depletion of the fatty-acid binding protein-3 (FABP3/MDGI) would induce LMP, we performed a thorough lipid composition analysis of the lysosomal membranes of control and MDGI-silenced cells. MDGI has been reported to efficiently bind the polyunsaturated fatty acids (PUFAs) (Richieri GV, Ogata RT, Zimmerman AW, Veerkamp JH, Kleinfeld AM. Fatty acid binding proteins from different tissues show distinct patterns of fatty acid interactions. *Biochemistry*. 2000;39:7197-204). Our lipid analyses show that MDGI silencing impaired trafficking of polyunsaturated fatty acids (FA) into cells resulting in significant alterations in the lipid composition of lysosomal membranes. Among the FAs preferred by MDGI is linoleic acid 18:2n-6, which cannot be synthesised de novo by the cells but has to be received from external sources. The fact that the PC and PE species (e.g. 36:2) harbouring this 18:2 FA chain were clearly decreased in the lysosomes of MDGI-silenced glioblastoma cells suggests that the impaired PUFA trafficking in these cells changed the lipid composition of lysosomal membranes. In line with this, the lysosomal membranes of MDGI-silenced cells contained larger proportions of monounsaturated FAs (MUFA) and less saturated FAs than the control lysosomes, which can be seen as fluidity compensation among these membrane bulk phospholipids. However, it should be recalled that MUFAs cannot replace the essential PUFAs in their specific biological functions, and it is likely that this compositional bias caused by MDGI silencing hampered lysosomal membrane integrity, dynamics, and vesicle secretion and changed the precursor pool of signalling molecules. In addition, the degree of unsaturation of the lysosomal ceramides (ratio of Cer 24:1 to Cer 24:0) markedly decreased upon MDGI silencing. Loss of the unsaturated Cer 24:1 species from the lysosomal membrane, leaves behind a membrane enriched with more rigid saturated 24:0, which probably made the membrane less elastic and prone to leakage. This novel finding explains the measured leakage of intra-lysosomal cathepsins into the tumour cell cytoplasm. These results have been added to the resubmitted manuscript and presented in Fig 5.*

All western blots should be repeated in order to load an equal amount of proteins. They should add a relative quantification of the protein of interest.

We have repeated all the Western blots to ensure equal loading of the proteins in each sample. We have also quantified the protein amounts relative to the loading control and marked the ratios in the figures of the resubmitted manuscript.

In Fig2 the authors should explain the meaning of the relative number of satellite tumors and how they measured it.

We apologize for the mistake and thank the Referee for pointing it out. Fig 2 does not show the relative number of secondary satellite tumours but the actual observed numbers. We have now corrected this in the Fig 2 of the resubmitted manuscript. Moreover, we chose to use a term "secondary tumour" throughout the manuscript instead of satellite tumour to describe independent neoplastic masses growing at a distance >300 µm from the primary tumour bulk.

To measure self-renewal ability of CSCs they should use methylcellulose assay and/or in vitro limiting dilution assay.

We have performed the methylcellulose assay for the control and MDGI-silenced patient-derived BT12 and BT13 cells, and quantified the size of the colonies after two weeks of culture. MDGI silencing significantly hampered the self-renewal ability of the glioblastoma CSCs. These results have been added as Fig 3B-C in the resubmitted manuscript. The colony formation assay in soft agar that assesses the aggressive, anchorage-independent growth has been transferred to the Supplementary Figures (Fig S1E-F).

For in vivo studies, they have used only one patient-derived cell line. Considering the high heterogeneity of GBM among patients, other tumor-derived cell lines should be used.

Due to the high heterogeneity of glioblastomas, we have now added in vivo results of two more patient-derived cell lines (BT13 and ZH305). The characterisation of their tumorigenicity revealed that the tested glioblastoma cells exhibit different growth patterns in murine brain following intracranial implantation. BT13 cells grow as non-invasive, angiogenic primary tumours, ZH305 cells grow highly diffusively without formation of tumour bulk, and BT12 cells show a mixed phenotype with primary tumour bulk and invasive growth. These results together confirm the vulnerability of the invasive glioblastoma cells to clemastine treatment. These results are shown in Fig 7 and Supplementary Fig S5B of the resubmitted manuscript.

They do not investigate in vivo for possible side effects of clemastine treatment.

The monitoring of the animal health and weight was followed during the experiments starting one day prior the tumour cell implantation and pursued until the euthanasia. No significant difference in the animal weights was observed. Following the brain samples collection, a macroscopy observation of the peripheral organs (kidneys, liver, stomach, spleen, heart, intestines, and lungs) was performed but no difference was found. As clemastine is metabolised by the liver and might also affect the renal, cardiovascular and splenic systems, each organ was weighted and normalised to the body mass. No significant differences were found between the vehicle and clemastine-treated groups. These results are shown in the Supplementary Fig S6 of the resubmitted manuscript. Eventually, no side effects apart from the transient drowsiness of the animals following drug injections was observed during the treatment. The maximal tolerated dose in mice is 730 mg/kg (data from the provider), which positions our selected dosage below that (one first dose of 100 mg/kg then 50 mg/kg, daily).

Discovered nearly 40 years ago, clemastine is FDA-approved and very well characterised molecule that is sold over-the-counter in many countries. It is currently used in the ReCOVER clinical trial aiming to treat patients suffering from lateral sclerosis.

The authors should repeat the evaluation of clemastine treatment in PDXs derived from the injection of MDGI-silenced cells.

We have now implanted both the control and MDGI-silenced BT12 cells intracranially into nude mice and performed the treatment with both the vehicle and clemastine. No tumour growth was observed when MDGI-silenced cells were implanted into the murine brain corroborating the results obtained in vitro and clemastine treatment did not change that. In addition, the control cells encoding the non-targeted scrambled shRNA grew very similarly to the parental BT12 cell line in vivo and clemastine treatment very efficiently eradicated the invasive glioblastoma cells. These results are shown in Fig 3E-I and Fig 7 in the resubmitted manuscript.

Referee #2 (Comments on Novelty/Model System for Author):

Authors have utilized very relevant patient-derived models.

Referee #2 (Remarks for Author):

In this manuscript Joncour et. al. have identified FABP3 as novel biomarker of glioma cell invasion. In addition, they also demonstrate that FABP3 silencing leads to glioma cell death via induction of LMP. Finally, using LMP inducing small molecule they demonstrated that LMP inducing drug significantly prolong survival of intracranial glioma bearing animals. This is extremely novel study with high translational impact, given that LMP inducing drug used in the study is already approved for use in humans. While authors have presented comprehensive data in support of their findings

some additional data will strengthen their findings and will improve the experience for avid scientific readers.

Suggestions:

Authors are requested to provide example IHC images representative of each staining score.

We have added the representative IHC micrographs that show the MDGI expression in clinical glioma samples (grades II-IV) as Fig 1A of the resubmitted manuscript. We also added sample of the epileptic brain that does not express MDGI as a negative control.

Authors are requested to look at the public databases (such as TCGA) and compare FABP3 expression level and survival of glioma patients.

This comment was also presented by Referee #3. Following the Referee's advice, we performed analyses of the MDGI/FABP3 mRNA expression in public databases. We used the GlioVis dataportal that contains data from gliomas collected from the TCGA and other datasets. These analyses showed similar results to our protein level analyses using glioma TMAs. High MDGI/FABP3 mRNA expression associated with poor survival in glioma patients (grades II-IV) but not in glioblastoma patients alone. It may be difficult to see significant differences in the glioblastoma patient survival since the overall survival rates are very short. In addition, the number of patients in the glioblastoma only cohort is smaller. Moreover, when the gene expression was analysed in the different anatomical structures of glioblastomas, MDGI/FABP3 was expressed at especially high levels in the leading edge of the tumours and in the infiltrating tumour cells. This is in accordance with the results we obtained with our glioblastoma cells both in vitro and in vivo. These results are presented in Fig 1C-D and in Supplementary Fig S1C of the resubmitted manuscript.

Authors are requested to look at the pharmacodynamics effects of Clemastine in intracranial xenografts, aka, do clemastine treated tumors in vivo demonstrate increase in LGALS1-puncta?

In response to this request by the Referee, we have stained the sections of xenografts with anti-galectin-1 (LGALS1) antibodies. As opposed to flat, single cells in culture, the compact tumour tissue made the interpretation of stainings a challenging task. However, it appears that clemastine treatment induced accumulation of galectin-1 to larger punctate structures compared to more diffusive staining in the vehicle treated xenografts. These punctate structures show strong staining for galectin-1, which appear brighter than in the control xenografts. Therefore, we prepared whole brain extracts from the vehicle and clemastine-treated xenografts and performed Western blot analyses of galectin-1. To be able to evaluate the amount of tumour cells in the brain and assess galectin-1 expression in those, we used human specific antibodies for both galectin-1 and vimentin. These results showed no significant differences in the galectin-1 levels between the groups suggesting that clemastine-treatment induced re-localization of galectin-1 also in vivo indicating induction of LMP. These results have been added as Supplementary Fig S5C-E in the resubmitted manuscript.

An experiment with cathepsin inhibitor to rescue LMP induced cell death downstream of FABP3 knockdown and clemastine treatment will comprehensively validate authors claim of MOA.

In order to investigate if cathepsin inhibition would be able to rescue the LMP induced by MDGI/FABP3 silencing, we have treated the control and MDGI-silenced BT12 and BT13 glioblastoma cells with a pan-cathepsin inhibitor K777. Indeed, the pan-cathepsin inhibitor was able to partially rescue the MDGI-silencing induced cell death. This result is shown as Fig 4L in the resubmitted manuscript.

Referee #3 (Remarks for Author):

The study by Le Joncour et al. investigated the roles of mammary-derived growth inhibitor (MDGI) and lysosomal membrane destabilization in gliomas. The study shows that MDGI expression correlates with survival in grades II-III gliomas but not in glioblastoma (GBM). MDGI expression did not correlate with glioma grade.

In our previous study we have shown that MDGI protein is expressed in a grade-dependent manner in human gliomas and its expression positively correlates with the histologic grade (Hyvönen et al., Novel target for peptide-based imaging and treatment of brain tumors. Mol Cancer Ther. 2014 Apr;13(4):996-1007). In the Supplementary Table S1 MDGI expression was compared between WHO grades II-III. MDGI is expressed at significantly lower levels in low grade gliomas compared to glioblastomas and although MDGI protein expression is higher in grade III gliomas compared to grade II gliomas no significant difference was observed.

The data presented in this manuscript are novel and potentially significant as very little is known about MDGI in gliomas and the use of LMP-triggering clemastine has not been tested as experimental GBM therapy before. The manuscript is clearly written. However, presented data are somewhat limited and superficial and the expression, functional and therapeutic data are disconnected. Substantial additional work is required to confirm and focus the overall message of the study.

Major critique:

The shown survival data are in lower grade gliomas, but all functional and therapeutic data are in GBM (there was no association with survival of MDGI in GBM). TCGA data should be analyzed for potential association with GBM survival or with glioma grade of MDGI. TCGA or other public data should also be analyzed for a potential differential expression of MDGI in GBM relative to either lower grade gliomas or normal brain.

This comment was also presented by Referee #2. As mentioned in our response to his/her comments, we performed analyses of the MDGI/FABP3 mRNA expression in public databases. We used the GlioVis datportal that contains data from gliomas collected from the TCGA and other datasets. These analyses showed similar results to our protein level analyses using glioma TMAs. High MDGI/FABP3 mRNA expression associated with poor survival in glioma patients (grades II-IV) but not in glioblastoma patients alone. It may be difficult to see significant differences in the glioblastoma patient survival since the overall survival rates are very short. In addition, the number of patients in the glioblastoma only cohort is smaller. Moreover, when the gene expression was analysed in the different anatomical structures of glioblastomas, MDGI/FABP3 was expressed at especially high levels in the leading edge of the tumours and in the infiltrating tumour cells. This is in accordance with the results we obtained with our glioblastoma cells both in vitro and in vivo. These results are presented in Fig 1C-D and in Supplementary Fig S1C of the resubmitted manuscript.

The overexpression of MDGI is done in one cell line only. The endpoint is invasion in vitro and in vivo. In vitro data in a second cell line should be generated. Also, why was tumor growth or animal survival not assessed in vivo? If MDGI is an important regulator of malignancy, its overexpression should affect in vivo xenograft growth and/or animal survival.

We have now overexpressed MDGI in the LN229 glioblastoma cells in addition to the U87MG cells. Also, in this cell line increased MDGI expression resulted in significantly increased number of invaded single cells and angiotropic tumours. In vivo results with U87MG and LN229 expressing MDGI-GFP and GFP controls are shown in Fig 2 of the resubmitted manuscript.

The U87MG cell line is known to form cyst-like neoplastic lesions when grafted in mice (Xie et al, 2015 EBioMedicine). This feature, shared by most of the glioma cell lines cultured with bovine serum supplement, generally raises many concerns about the relevance of its use as a pre-clinical model of glioblastoma. The mechanistic approach, to show that manipulation of MDGI expression in a non-invasive model could redirect it towards a more invasive phenotype was the rationale of selecting U87MG for our experiments. Although a significant effect of the tumorigenesis was observed intra-cranially, the animals from both groups eventually started to show signs of sickness simultaneously, e.g. hemiplegia due to intracranial pressure increase caused by the growth of the bulky primary tumour. Therefore, the endpoint was the same for both cohorts.

How does MDGI regulate LMP? This issue is mentioned in the discussion but not experimentally addressed. Mechanistic experiments that uncover the mode of action of MDGI would strengthen the findings.

*This comment was also presented by Referee #1. To understand how depletion of the fatty-acid binding protein-3 (FABP3/MDGI) would induce LMP, we performed a thorough lipid composition analysis of the lysosomal membranes of control and MDGI-silenced cells. MDGI has been reported to efficiently bind the polyunsaturated fatty acids (PUFAs) (Richieri GV, Ogata RT, Zimmerman AW, Veerkamp JH, Kleinfeld AM. Fatty acid binding proteins from different tissues show distinct patterns of fatty acid interactions. *Biochemistry*. 2000;39:7197-204). Our lipid analyses show that MDGI silencing impaired trafficking of polyunsaturated fatty acids (FA) into cells resulting in significant alterations in the lipid composition of lysosomal membranes. Among the FAs preferred by MDGI is linoleic acid 18:2n-6, which cannot be synthesised de novo by the cells but has to be received from external sources. The fact that the PC and PE species (e.g. 36:2) harbouring this 18:2 FA chain were clearly decreased in the lysosomes of MDGI-silenced glioblastoma cells suggests that the impaired PUFA trafficking in these cells changed the lipid composition of lysosomal membranes. In line with this, the lysosomal membranes of MDGI-silenced cells contained larger proportions of monounsaturated FAs (MUFA) and less saturated FAs than the control lysosomes, which can be seen as fluidity compensation among these membrane bulk phospholipids. However, it should be recalled that MUFAs cannot replace the essential PUFAs in their specific biological functions, and it is likely that this compositional bias caused by MDGI silencing hampered lysosomal membrane integrity, dynamics, and vesicle secretion and changed the precursor pool of signalling molecules. In addition, the degree of unsaturation of the lysosomal ceramides (ratio of Cer 24:1 to Cer 24:0) markedly decreased upon MDGI silencing. Loss of the unsaturated Cer 24:1 species from the lysosomal membrane, leaves behind a membrane enriched with more rigid saturated 24:0, which probably made the membrane less elastic and prone to leakage. This novel finding explains the measured leakage of intra-lysosomal cathepsins into the tumour cell cytoplasm. These results have been added to the resubmitted manuscript and presented in Fig 5.*

The clemastine data are not directly related to MDGI. Is the clemastine effect mediated by MDGI inhibition? That would be worthy of investigating. If not, the effect of MDGI silencing on in vivo tumor growth should be tested.

As suggested by the Referee, we tested whether clemastine-treatment affects the MDGI expression in the glioblastoma cells. No effect in MDGI expression could be observed when BT12, BT13 or ZH305 cells were treated with clemastine for 24 hrs. Therefore, it seems that both MDGI silencing and clemastine-treatment induce LMP but MDGI is not downstream of clemastine in this event. These results have been added as Supplementary Fig S5A in the resubmitted manuscript. We have also now implanted the control and MDGI-silenced cells intracranially in nude mice. The control shRNA expressing BT12 cells grew very similarly to the parental BT12 cells, while the MDGI-silenced cells failed to form tumours in vivo. These results are presented as Fig 3E-I in the resubmitted manuscript.

Minor critique:

- In which cell line(s) were the data shown in **figure 4 performed**. This is not mentioned in the text or in the legend.

These experiments were performed in the BT12 patient-derived glioblastoma cells. This has now been added in the Results section (page 7) and in the legend of Fig 4 of the resubmitted manuscript.

- "**Instabilization**" is not an English word. "**Destabilization**" or "instability" should be used instead.

We have changed the word instabilization with destabilization.

2nd Editorial Decision

13 December 2018

Thank you for the resubmission of your manuscript to EMBO Molecular Medicine. We have now heard back from the three referees whom we asked to evaluate your manuscript:

Referees 1 and 2, who had reviewed the manuscript before, are overall positive and support publication of the article in EMBO Molecular Medicine pending statistical analysis of figure 5 (referee 2, comment #2). Referee 3 reviewed the manuscript for the first time, and while also being

supportive of publication, requires minor additional experiments (expression of MDGI in different GBM subtypes, Western blots of HIF1a). Text clarifications and improved discussion will also be necessary. Experiments addressing caspase 3 activity (reviewer 3, comment #5) would be welcome, but are not required for acceptance of the manuscript.

EMBO Molecular Medicine encourages a single round of revision only and therefore, acceptance or rejection of the manuscript will depend on the completeness of your responses included in the next, final version of the manuscript.

EMBO Molecular Medicine has a "scooping protection" policy, whereby similar findings that are published by others during review or revision are not a criterion for rejection. Should you decide to submit a revised version, I do ask that you get in touch after three months if you have not completed it, to update us on the status. Please also contact us as soon as possible if similar work is published elsewhere. If other work is published, we may not be able to extend the revision period beyond three months.

I look forward to receiving your revised manuscript.

***** Reviewer's comments *****

Referee #2 (Comments on Novelty/Model System for Author):

Models used for the study are adequate. Authors have utilized patient-derived cell lines of GBM which have been a well-accepted model systems in the research community.

Referee #2 (Remarks for Author):

This version of the manuscript is a significant improvement to the previous version of the manuscript. Authors have satisfactorily addressed reviewer critiques. In my opinion, this manuscript is acceptable as is.

Referee #3 (Remarks for Author):

The revised manuscript by Le Joncour et al. satisfactorily addresses most critique of this reviewer. Among other, the study added experiments using a second cell line, performed in vivo experiments, performed lipid mass spectrometry to uncover the mechanism of action of MDGI and experimentally clarified the effects of clemastine on MDGI. As a result, the manuscript is substantially improved. The following issues remain:

- The lipid composition data that were added contribute to the understanding of the mechanism of action of MDGI in gliomas, but the data are somewhat descriptive.
- The lysosomal membrane lipid composition data of figure 5 have not been statistically analyzed. Was the experiment only performed once? If yes, it should be repeated and statistical analyses should be performed.
- Since the clemastine effects are not mediated by MDGI, the clemastine data are not directly related to the rest of the findings.

Referee #4 (Remarks for Author):

In this manuscript by Le Joncour et al., the authors have identified lysosome membrane permeabilization (LMP) as the cell death mechanism induced by MDGI/FABP3 silencing in glioblastoma cells. They show that MDGI silencing leads to change in the composition of lysosome

membranes with consequent leakage of proteolytic enzymes to the cytoplasm and cell death. In vivo, MDGI overexpression leads to increase formation of secondary tumors to distant sites, while silencing in the patient-derived spheroids BT12 completely suppress tumor formation. The authors suggest that GBM might be sensitive to LMP-mediated cell death and test the use of LMP-inducing agent to possibly inhibit GBM cells growth. Interestingly, clemastine strongly affects growth of a very invasive patient-derived spheroids (ZH305) indicating that it could be a valuable option for GBM treatment.

The study is very interesting and presents novel data with a strong potential to improve GBM treatment. The manuscript is well written and the results are clearly presented. The authors have addressed all the comments suggested by the reviewers in an exhaustive manner. However, a few points are not clearly explained or discussed and should be further addressed to make manuscript more complete.

Major comments

- 1) The difference between "Glioma specific survival" (Fig. S1A) and "overall survival" (Fig. 1B) is not clear. According to the explanation in the text Fig. 1B should refer to the overall survival analysis of all glioma patients (low and high grade) but this is not clearly stated. Moreover, it appears that "overall" is used to indicate "all gliomas". Similarly, it should be clearly indicated that Fig. S1B refers to "overall survival of GBM patients".
- 2) The authors comment on the fact that no difference in "overall GBM patients survival" is observed, suggesting that this could be due to the short survival of GBM patients. However, although the prognosis of GBM patients is usually very bad, difference in survival are due to IDH1 mutation and consequent G-CIMP phenotype. It should be examined the expression of MDGI in different GBM subtypes (mes, pron, class) or G-CIMP+/- . If MDGI is associated with survival due to the intrinsic survival difference associated with tumor grade then no difference in MDGI expression in GBM subtypes should be observed.
- 3) In Fig. 1G a western blot showing HIF1 α expression should be included.
- 4) The authors show that MDGI is induced by hypoxia but these results are not clearly connected to the rest of the story. The authors should consider discussing these results in the context of the other observations (i.e. MDGI expression in different regions and tumor phenotype in vivo).
- 5) The data regarding caspase 3 are not very clear. In Fig. S3A, the authors show a clear reduction of caspase 3 cleavage at day 6 but then they conclude that observed cell death is caspase-independent based on the result of the antibody array of apoptosis-associated proteins (Fig. S3B). However, increased calpain activity upon MDGI/FABP3 silencing has been previously observed (Shen et al., Cell Biochemistry and Biophysics). The connection of MDGI/FABP3 with caspase 3 activation and apoptosis based on the author's and previous results should be discussed. Considering that LMP has been reported to occur in presence or absence of caspase 3 activation (Juai et al., 2013 Journal of Cell Science; Boya et al., 2003 JEM) the role of caspase 3 should be more clearly addressed, i.e. by using a caspase 3 inhibitor and measuring calpain 3 activity.

Minor comments

- 1) At page 7, it would be better to say that "MDGI has been shown to affect EGFR trafficking" instead of "to interact with EGFR" since it better reflects the content of the cited paper.
- 2) In Fig. S6, the "y axis" legend is missing in panel A while there is no indication of which body part is measured in panels B-E.
- 3) In the supplementary methods, there is a repetition in the description of the commercial cell lines.
- 4) Not all the primary and commercial cell lines are included in the cell lines description in the methods.
- 5) The number of the animals used for the intracranial injection is not consistent between the methods and supplementary methods sections. Moreover, it is not mentioned how many animals for the injection of BT13 and ZH305 cells were used.

***** Reviewer's comments *****

Referee #2 (Comments on Novelty/Model System for Author):

Models used for the study are adequate. Authors have utilized patient-derived cell lines of GBM which have been a wellaccepted model systems in the research community.

Referee #2 (Remarks for Author):

This version of the manuscript is a significant improvement to the previous version of the manuscript. Authors have satisfactorily addressed reviewer critiques. In my opinion, this manuscript is acceptable as is.

We thank the referee for the positive feedback on our manuscript.

Referee #3 (Remarks for Author):

The revised manuscript by Le Joncour et al. satisfactorily addresses most critique of this reviewer. Among other, the study added experiments using a second cell line, performed in vivo experiments, performed lipid mass spectrometry to uncover the mechanism of action of MDGI and experimentally clarified the effects of clemastine on MDGI. As a result, the manuscript is substantially improved. The following issues remain:

- The lysosomal membrane lipid composition data of figure 5 have not been statistically analyzed. Was the experiment only performed once? If yes, it should be repeated and statistical analyses should be performed

The lysosomal damage induced by MDGI-silencing that we have thoroughly investigated made the isolation of lysosomal membranes a particularly complex task. It was a great challenge to obtain the required number of lysosomes to perform the mass spectrometric analyses of the lipid content, since MDGI silencing decreased the number of lysosomes as a consequence of the LMP. Therefore, both the culture of patient-derived spheroids and lentivirus production have been adjusted to large-scale production. Thus, one readout actually constitutes of six independent spheroid culture samples pooled together.

As requested by the Referee, we have now repeated the analyses. The obtained results are very similar to the previous ones. Please see the figure below. Despite the fact that the one analysed sample is a combination from several independent samples, we now have two biological replicate readouts/cell line, which does not allow the statistical analyses. Therefore, we show the results from one experiment without the statistical analyses (Figure 5 of the revised manuscript). We can add the results from the second experiments as an Appendix Figure in case the Referee and/or Editor consider it appropriate.

Experiment 2. January 2019.

- Since the clemastine effects are not mediated by MDGI, the clemastine data are not directly related to the rest of the findings.

Referee is absolute correct to point out that the clemastine data is not directly related to the data obtained with MDGI. However, our purpose was to study whether this vulnerability of glioblastoma cells to lysosomal membrane destabilisation, which we discovered by silencing MDGI, could be induced by any drug molecule that could be used for the pre-clinical studies. Thus, destabilisation of the lysosomal membranes of glioblastoma cells can be induced either by MDGI silencing or by an antihistamine drug, and independently of the trigger, LMP leads to death of glioblastoma cells. Therefore, LMP induction provides a possible mechanism to target glioblastoma cells for destruction.

Referee #4 (Remarks for Author):

In this manuscript by Le Joncour et al., the authors have identified lysosome membrane permeabilization (LMP) as the cell death mechanism induced by MDGI/FABP3 silencing in glioblastoma cells. They show that MDGI silencing leads to change in the composition of lysosome membranes with consequent leakage of proteolytic enzymes to the cytoplasm and cell death. In vivo, MDGI overexpression leads to increase formation of secondary tumors to distant sites, while

silencing in the patient-derived spheroids BT12 completely suppress tumor formation. The authors suggest that GBM might be sensitive to LMP-mediated cell death and test the use of LMP-inducing agent to possibly inhibit GBM cells growth. Interestingly, clemastine strongly affects growth of a very invasive patient-derived spheroids (ZH305) indicating that it could be a valuable option for GBM treatment.

The study is very interesting and presents novel data with a strong potential to improve GBM treatment. The manuscript is well written and the results are clearly presented. The authors have addressed all the comments suggested by the reviewers in an exhaustive manner. However, a few points are not clearly explained or discussed and should be further addressed to make manuscript more complete.

Major comments

1) The difference between "Glioma specific survival" (Fig. S1A) and "overall survival" (Fig.1B) is not clear. According to the **explanation in the text Fig.1B should refer to the overall survival analysis of all glioma patients** (low and high grade) but this is not clearly stated. Moreover, it appears that "overall" is used to indicate "all gliomas". Similarly, it should be clearly indicated that **Fig.S1B** refers to "overall survival of GBM patients".

We apologise the unclear presentation of the analysed samples. Fig 1B shows the overall survival of lower grade (II and III) glioma patients while Fig S1A shows the glioma-specific survival of lower grade (II and III) glioma patients and Fig S1B shows the survival curves for glioblastoma (grade IV) patients only. We have now clarified this in the main text (last paragraph, page 4) and in the legends for Figs 1B and S1A-B in the revised manuscript.

2) The authors comment on the fact that no difference in "overall GBM patients' survival" is observed, suggesting that this could be due to the short survival of GBM patients. However, although the prognosis of GBM patients is usually very bad, difference in survival are due to **IDH1 mutation and consequent G-CIMP phenotype**. It should be examined the expression of **MDGI in different GBM subtypes** (mes, pron, class) or **G-CIMP+/-**. If MDGI is associated with survival due to the intrinsic survival difference associated with tumor grade then no difference in MDGI expression in GBM subtypes should be observed.

We have now further analysed MDGI expression using the GlioVis data portal (the TCGA GBM, LGG, and GBMLGG datasets). We analysed MDGI expression in the different histological glioma subclasses (grades II-IV) and different glioblastoma (grade IV only) subtypes as well as the association of MDGI expression with the IDH mutant and G-CIMP status. We describe these results in the main text (page 5, second paragraph) and show them as Figs 1D and S1D-F in the revised manuscript.

3) In Fig. 1G a western blot showing **HIF1 α** expression should be included.

We have added the HIF1 α expression in the studied cell lines in Fig 1H as requested by the Referee.

4) The authors show that MDGI is induced by hypoxia but these results are not clearly connected to the rest of the story. The authors should consider discussing these results in the context of the other observations (i.e. MDGI expression in **different regions and tumor phenotype in vivo**).

MDGI induction by hypoxia is indeed not in the centre focus of our manuscript. MDGI has been reported to be induced by hypoxia in a HIF-1 α -dependent manner in U87MG cells (Bensaad et al., Fatty acid uptake and lipid storage induced by HIF-1 α contribute to cell growth and survival after hypoxia-reoxygenation, Cell Reports 9:39-365, 2014). Our previous paper showed that MDGI is expressed in human brain tumours in a grade-dependent manner and its expression positively correlates with the histologic grade of the tumour. In clinical glioblastoma samples MDGI expression was concentrated in the peri-necrotic hypoxic areas of the tumours (Hyvönen et al., Novel target for peptide-based imaging and treatment of brain tumors, Molecular Cancer Therapeutics 13:996-1007, 2014). Therefore, we evaluated the MDGI induction by hypoxia experimentally in this manuscript.

5) The data **regarding caspase 3** are not very clear. In Fig. S3A, the authors show a clear reduction

of caspase 3 cleavage at day 6 but then they conclude that observed cell death is caspase-independent based on the result of the antibody array of apoptosis-associated proteins (Fig. S3B). However, increased calpain activity upon MDGI/FABP3 silencing has been previously observed (Shen et al., *Cell Biochemistry and Biophysics*). The connection of MDGI/FABP3 with caspase 3 activation and apoptosis based on the author's and previous results should be discussed. Considering that LMP has been reported to occur in presence or absence of caspase 3 activation (Huai et al., 2013 *Journal of Cell Science*; Boya et al., 2003 *JEM*) **the role of caspase 3 should be more clearly addressed**, i.e. by using a caspase 3 inhibitor and measuring calpain 3 activity.

According to the literature LMP can be induced by different means and it may be either caspase-dependent or - independent. In the paper by Shen et al. (Silencing of FABP3 inhibits proliferation and promotes apoptosis in embryonic carcinoma cells, Cell Biochemistry and Biophysics, 66:139-146, 2013) the authors studied the effect of FABP3/MDGI overexpression and silencing on differentiation of embryonic myocardial cells using the P19 embryonic carcinoma cells. In their model both overexpression and silencing of FABP3/MDGI induced apoptosis of P19 cells. However, FABP3/MDGI silencing did not affect the DMSO-induced differentiation of the cells. Their model significantly differs from ours in several aspects. They grow the cells in the presence of serum and use serum withdrawal to induce apoptosis. In this setup Shen et al. showed that caspase-3 activity was increased in FABP3/MDGI silenced cells. Boya et al. (Lysosomal membrane permeabilisation induces cell death in a mitochondrion-dependent fashion, The Journal of Experimental Medicine, 197:1323-1334, 2003) studied in detail LMP induction by using two quinolone antibiotics (ciprofloxacin, CP and norfloxacin, NFX). NFX induces LMP only when used in combination with UV light. CPX and NFX triggered LMP in caspase-independent LMP. The authors also showed that in their study mitochondrial membrane permeabilisation occurred downstream of LMP. These studies were performed using stably transfected HeLa cells and Rat-1/myc fibroblasts, and SV40-transfected murine embryonic fibroblasts (MEFs). In the paper by Huai et al. (TNF α -induced lysosomal membrane permeability is downstream of MOMP and triggered by caspase-mediated NDUF51 cleavage and ROS formation. Journal of Cell Science, 126:4015-4025, 2013) the authors studied TNF α +cycloheximide-induced lysosomal membrane permeabilisation using wt and SV40-transfected murine embryonic fibroblasts (MEFs) as well as HeLa cells. Their results show that TNF α -induced LMP is caspase-dependent and downstream of mitochondrial outer membrane permeabilisation, and that LMP is not an initiating process but occurs rather as an amplification loop downstream of MOMP. Taken together, it seems that cell death induced by LMP is context-dependent and can involve caspase activation and may occur up- or downstream of mitochondrial membrane permeabilisation.

In our patient-derived glioblastoma cells we did not observe activation of caspase-3 (increase in the cleaved form) when we studied the levels of both total caspase-3 and its cleaved activated form by Western blot, instead we detected decrease in the amount of both activated caspase-3 and total caspase-3 (Appendix Fig S3A). When we studied the levels of apoptosis-associated proteins by using antibody array, we also detected decrease in the amount of cleaved active caspase-3 (Appendix Fig S3C and D). Since the fold-change (0.61) exceeded the threshold (< 0.60) it was considered as unchanged in the Appendix Fig S3C and D. This sentence has now been added to the revised manuscript (page 8, first paragraph). In addition, we detected significant increase in the cytoplasmic cathepsin B activity and partial rescue of viability of the MDGI-silenced cells (Fig 4K and L). These results suggest that FABP3/MDGI-silencing in the patient-derived glioblastoma cells leads to LMP that involved cathepsins and was most probably independent of the caspase activity. Our results also show that the normal cells are more resistant to LMP-induction by clemastine than glioblastoma cells (Fig 6). However, we have not studied the downstream effectors in more detail since it is not the focus of our manuscript. We have added the references mentioned by the Referee and a sentence about this in the Discussion of the revised manuscript (page 13, second paragraph) "LMP is an intracellular cell death pathway that can be either caspase-independent or caspase-dependent and it can occur either up- or downstream of mitochondrial membrane permeabilisation depending on the cell type and/or LMP-inducer (Aits & Jaattela, 2013, Boya et al., 2003, Huai et al., 2013, Shen et al., 2013)."

Minor comments

1) At page 7, it would be better to say that "MDGI has been shown to affect EGFR trafficking" instead of "to interact with EGFR" since it better reflects the content of the cited paper.

We have now changed the wording as suggested by the Referee.

2) In Fig.S6, the "y axis" legend is missing in panel A while there is no indication of which body part is measured in panels B-E.

We thank the Referee for pointing this out. We have added the title for the y-axis (Animal weight (g)) and named the organs in the panels. These results are now shown as Appendix Fig S4B-F in the revised manuscript.

3) In the supplementary methods, there is a repetition in the description of the commercial cell lines.

We have deleted the repetition in the revised Supplementary methods.

4) Not all the primary and commercial cell lines are included in the cell lines description in the methods.

We carefully checked all cell lines used and added the missing ones in the methods.

5) The number of the animals used for the intracranial injection is not consistent between the methods and supplementary methods sections. Moreover, it is not mentioned how many animals for the injection of BT13 and ZH305 cells were used.

We apologise the missing information and have now added these in the revised manuscript.

3rd Editorial Decision

21 March 2019

Thank you for the submission of your revised manuscript to EMBO Molecular Medicine. We have received the enclosed reports from the referees, who are now supportive of publication. I am therefore pleased to inform you that we will be able to accept your manuscript once final editorial amendments have been completed.

***** Reviewer's comments *****

Referee #3 (Remarks for Author):

The authors satisfactorily addressed the previous critique. The manuscript is improved.

Referee #4 (Comments on Novelty/Model System for Author):

The use of patient-derived GBM cell lines is considered to suitable model systems in the brain tumor field.

Referee #4 (Remarks for Author):

The authors have addressed all the comments in an exhaustive way. The data discussion is improved. In my opinion the paper is acceptable for publication.

3rd Revision - authors' response

3 April 2019

Authors made the requested editorial changes.

Corresponding Author Name: Pirjo LAAKKONEN

Manuscript Number: EMM-2018-09034-V2